

# Integrability of open boundary driven quantum circuits

**Chiara Paletta[1] and Tomaž Prosen[1,2]**

**1** Department of Physics, Faculty of Mathematics and Physics,
University of Ljubljana, Jadranska 19, SI-1000 Ljubljana, Slovenia
**2** Institute of Mathematics, Physics and Mechanics,
Jadranska 19, SI-1000, Ljubljana, Slovenia

## Abstract

In this paper, we address the problem of Yang-Baxter integrability of doubled quantum circuit of qubits (spins 1/2) with open boundary conditions where the two circuit replicas are only coupled at the left or right boundary. We investigate the cases where the bulk is given by elementary six vertex unitary gates of either the free fermionic XX type or interacting XXZ type. By using the Sklyanin's construction of reflection algebra, we obtain the most general solutions of the boundary Yang-Baxter equation for such a setup. We use this solution to build, from the transfer matrix formalism, integrable circuits with two step discrete time Floquet (aka brickwork) dynamics. We prove that, *only* if the bulk is a free-model, the boundary matrices are in general non-factorizable, and for particular choice of free parameters yield non-trivial unitary dynamics with boundary interaction between the two chains. Then, we consider the limit of continuous time evolution and we give the interpretation of a restricted set of the boundary terms in the Lindbladian setting. Specifically, for a particular choice of free parameters, the solutions correspond to an open quantum system dynamics with the source terms representing injecting or removing particles from the boundary of the spin chain.


doi:10.21468/SciPostPhys.18.1.027

# 1 Introduction

The study of the general dynamical behaviour of complex non-linear interacting systems is of interest in many area of physics, including condensed matter theory and AMO physics, statistical physics, (quantum) field theory, string theory and quantum information theory.

**Quantum integrable models** Having an exactly solved model offers a significant advantage, as it often characterizes the universal behaviour of a broader class of potentially unsolvable models, thereby providing the most precise understanding of physical reality. What characterizes the *solvability* in certain models is the presence of numerous conserved quantities vastly restricting their dynamics and allowing for exact solutions. While this is expected in isolated systems of non-interacting particles, it is remarkable that this phenomenon can also occur in certain interacting theories. The property of having enough – a complete set of – *conserved quantities* is one of the possible definitions of a quantum *integrable* model. However, there are various possible precise definitions of integrability [1], and our main focus in this work is on *Yang-Baxter integrable* models, in particular on Yang-Baxter integrable quantum spin chains. Those are characterized by an *R*-matrix solution of the Yang-Baxter equation (YBE) generating the system's time evolution.

The physical interpretation of YBE is the factorization property of scattering, meaning that any scattering of three (or generally *n*) particles among each other can be decomposed into a sequence of scatterings between pairs of particles, irrespective of their order.

The *R*-matrix (in appropriate representation) can be used as a building block of the *transfer matrix*, an object used to construct all the relevant conserved charges characterizing the model. This commutation properties of the transfer matrix, together with the YBE, is the cornerstone of the algebraic Bethe Ansatz [2], a method for diagonalising integrable many-body Hamiltonians. This exact technique provides also insightful information on the calculation of correlation functions. When the spin chain has open boundary conditions, to define integrability, together with the standard YBE, two extra relations need to be considered, [3–5]. Those describe, respectively, the scattering of particles with the right and left boundaries. The solutions of these relations are referred to as *reflection matrices* (or *K*-matrices). The technique of algebraic Bethe Ansatz generally extends for this class of model, see e.g. Ref. [6].

Considering the versatility of integrable models, it is important to note that they have been extensively analyzed across various fields of theoretical physics. Specifically, in the last decade they found prominent application in the context of *quantum computation* and *quantum simulation*, where the so-called quantum circuits represent universal discrete-time models of many-body quantum dynamics. Our paper will mainly be concerned with integrability of a certain class of quantum circuits which can either be mapped to Hamiltonian evolution or open system dynamics, or be considered as integrable models of quantum computation in its own right.

**Quantum circuits**    A many body operator *M* will be referred to as a quantum circuit when it can be factorized into a finite sequence of 2-particle operators, the so-called gates. An *integrable quantum circuit* may be realized when *M* commutes with a transfer matrix of an integrable model. Based on this idea, the *integrable trotterization* procedure was developed. The idea originates from a work by Baxter [7], where it was understood that the values of the so-called inhomogeneities of the transfer matrix of the six-vertex model can be fixed to define a discrete time parallel update on a periodic lattice. This idea has been elaborated in [8,9], by providing a general framework (independent of the *R*-matrix considered) to define an integrable unitary discrete time evolution. The trotterization procedure works with either periodic and open boundary conditions and in both cases it appears to be two-step *Floquet* (time-periodic) dynamics.

The advantage of having an integrable quantum circuit relies on the possibility to use the integrability technique to compute the spectrum. Those analytical results may be used for the calibration and error mitigations in modern engineered quantum platforms, [10,11]. The setup of integrable quantum circuits has also been instrumental for demonstration of universal superdiffusive scaling aspects of quantum dynamics, [12]. Furthermore, in a particular limit, the connection between the Floquet construction and a non-rational conformal field theory was established, [13]. It was also shown that for the quantum circuit constructed from the trotterization of the XXZ spin chain, strong zero modes can be constructed in certain regions of parameter space, [14]. Recently, in the setup of integrable quantum circuits, correlation functions of strings of spin operators were computed, [15].

**Open quantum circuit**    An interesting class of quantum circuits that we discuss in this work, is the one where the quantum gates operate on density operators directly rather than on the pure quantum states. Using the vectorization (aka thermofield double) representation, we can consider gates which act on the tensor products of pairs of local Hilbert spaces. A general Krauss representation of time evolution of a density matrix then results in a non-unitary quantum circuit. In Ref. [16] it was demonstrated that trotterization of the Hubbard model

with imaginary interaction strength corresponds to an integrable open quantum circuit with a dephasing noise. This is an example of the general mapping between Liouvillians of open many-body systems and Bethe-Ansatz integrable systems on (thermofield) doubled Hilbert spaces [17–20].

The quantum gate that we consider in this work is given by the tensor product of two elementary quantum gates. This corresponds, in the continuum time evolution, to the coherent evolution part of the *Lindblad master equation*, [21, 22].

**Integrability in boundary driven diffusive Lindbladian systems**    Placing a quantum spin chain (system) in contact with a Markovian environment at the boundaries results in a driven diffusive system. The dynamic is governed by the Lindblad master equation, explicitly

$$\dot{\rho}(t) = \frac{d}{dt}\rho = \mathcal{L}\rho := i[\rho, H] + \Gamma \sum_j \Big[ \ell_j \rho \ell_j^\dagger - \frac{1}{2}\{\ell_j^\dagger \ell_j, \rho\} \Big], \tag{1}$$

where $H$ is the Hamiltonian of the system, $\Gamma$ is the strength of the coupling between the system and the environment, $\ell_j$ are the jump operators and describe the effective action of the environment on the system. In a boundary driven setup, we assume that the jump operators $\ell_j$ are all local and supported at the system left or right boundary.

The stationary state of this model is the so-called *non equilibrium steady state* (NESS) and satisfies $\mathcal{L}\rho_{\text{NESS}} = 0$. In 2011, the NESS density operator of the boundary driven Lindblad master equation for the XXZ spin 1/2 chain has been solved in terms of *matrix product operator* Ansatz for particular spin source/sink boundary operators, [23, 24]. For a review see [25]. Initially, these solutions appeared unrelated to the conventional theory of quantum integrability. However, later it was understood that they are connected to the infinite-dimensional solutions of YBE associated with non-unitary irreducible representations of the model's quantum group symmetry, [26–28]. Specifically, a link was found between the matrix product form of the non-equilibrium steady states and the integrable structure of the bulk Hamiltonian. For a quantum XX chain in the presence of bulk dephasing and arbitrary local boundary spin driving, it is possible to write the solutions for the NESS to the 2nd order in the driving strength of [29]. For the XX spin chain of finite length coupled to reservoirs at both ends, the NESS can be written as an MPO with fixed bound dimension 4 independent of the chain length, [30]. Another class of models for which, moreover, the full spectrum and eigenvectors were calculated is the XY spin chain with jump operator being linear in canonical fermionic operators, [31].

**Motivation of the paper**    The initial motivation for this paper was to explore any potential connection between the matrix product form of the NESS and the Yang-Baxter integrability structures of the entire spin chain (boundary+bulk). More specifically, a Lindblad system is considered Yang-Baxter integrable if the generator of the dynamics $\mathcal{L}$ commutes with an infinite number of conserved superoperators generated by a transfer matrix, [17–20]. In particular, the bulk dynamics governed by the Hamiltonian is related to an *R*-matrix that solves the Yang-Baxter equation, while the boundary terms correspond to *K*-matrices that solve the boundary Yang-Baxter equations. We consider the case where the Hamiltonian is the sum of range 2 operators, $H = \sum_{i=1}^L h_{i,i+1}$. As we clarify in sec. 5, it is convenient to express the evolution of the density matrix in a doubled Hilbert space. The bulk evolution corresponds to $i(1 \otimes H^T - H \otimes 1)$. The *R*-matrix associated to it is $R'(u)|_{u=0} \sim P(1 \otimes h^T - h \otimes 1)$. This motivates us to consider an elementary gate with factorized form $R(u) = r(u) \otimes r(-u)$, where $r(u)$ is the *R*-matrix of a spin 1/2 chain, related to the Hamiltonian density as $r'(u)|_{u=0} = p\,h$. We study the boundary reflection algebra $K$ associated with this $R$ matrix, focusing on whether all solutions take a factorized form or not. The first case, where the solution is factorizable, is less interesting as it corresponds to two uncoupled quantum circuits (or spin chains in the

continuum time limit), while the second case, which is non-factorizable, is non-trivial. For the non-factorizable solution, we analyze the conditions under which it can be expressed as the dissipator term of a Lindbladian (1), and we compare these results with the existing literature on NESS.

## 1.1 Short summary of the paper

This paper is organized as follows. In section 2, we review key concepts of quantum integrability for spin chains with both periodic [2] and open boundary conditions. For the case with open boundary condition, we detail the Sklyanin construction [3] for the left and right reflection algebras and summarize the protocol for constructing integrable quantum circuits with two-step Floquet dynamics, [8,9]. From section 3, the original part of the paper starts. Initially, we define the two-replica quantum circuits coupled through the boundary and explain the method for solving the boundary Yang-Baxter equations. The quantum elementary gate of the bulk is given by the tensor product of pairs two qubit gates. A central question we investigate is whether the quantum gates corresponding to the left and right boundaries exhibit a non-factorized expression. In section 4, we report our results. We begin by examining several integrable models: interacting models such as the Heisenberg XXX and XXZ spin chains, as well as the free fermion XX spin chain. For each model, we discuss the solution of the Sklyanin reflection algebra in the doubled (two replica) Hilbert space. Our main finding is that the solution of the reflection algebra is non-factorized only in the case where the bulk is the free fermion XX spin chain. In that case, the boundary interaction can be non-trivial and the model, restricting the possible values of the free parameters, may represent an interesting example of integrable unitary quantum dynamics in its own right. In section 5, we then discuss the implications of our results for continuous time and open system dynamics. The conserved charges of the models are derived from the higher-order derivatives of the transfer matrix. We analyze under which conditions the boundary terms can be expressed as a Lindbladian evolution. We conclude that this identification is only possible if the operators at the left and right boundaries of the spin chain act as sources of particle injection or removal. This implies that the Yang-Baxter integrability property of a model (both in the bulk and in the boundaries) imposes stricter conditions than those required for finding the analytical expression of the NESS of the model. While finding the NESS is possible for a broader class of models [24], the Yang-Baxter integrability property is more restrictive.

# 2 Set up: Building an integrable quantum circuit

In section 2.1, we review some key concepts about quantum integrability for both periodic and open boundary conditions. In section 2.2, we summarize the well known protocol for constructing integrable quantum circuits with two-step Floquet dynamics.

## 2.1 Quantum integrability in a nutshell

### 2.1.1 Periodic boundary condition

A quantum integrable model has an infinite number of conserved charges and it is characterized by an $R$-matrix solution of the **Yang-Baxter equation** (YBE), [7]

$$R_{12}(u)R_{13}(u+v)R_{23}(v) = R_{23}(v)R_{13}(u+v)R_{12}(u).\qquad(2)$$

This is a matrix relation defined in the triple tensor product vector space $\mathrm{End}(V \otimes V \otimes V)$, with $V \equiv \mathbb{C}^n$ the $n$-th dimensional complex vector space. The $R$-matrix is defined in $\mathrm{End}(V \otimes V)$

and in the indexed version $R_{ij}$, the subscripts denote which of the three spaces $R$ acts on nontrivially (for example $R_{12} = R \otimes \mathbb{I}$, $\mathbb{I}$ being the identity operator in $\mathbb{C}^n$) and $u, v$ are known as spectral parameters and can take values in $\mathbb{C}$. At both classical[1] and quantum levels, the YBE is considered the crucial component of integrability.

We can associate to the $R$-matrix an algebra defined by the so called RTT relation

$$R_{12}(u_1 - u_2)T_1(u_1)T_2(u_2) = T_2(u_2)T_1(u_1)R_{12}(u_1 - u_2). \tag{3}$$

This relation is defined in $\text{End}(W_1 \otimes W_2 \otimes V^{\otimes L})$, where $W_i$ are auxiliary spaces and $V$ is the physical (quantum) space and $L$ is the length of the lattice. In what follows, we always consider the fundamental representation, where $W_1$ and $W_2$ are isomorphic to $V$. In this case, $T(u)$ can be constructed from the $R$-matrix,

$$T_0(u, \mathbf{w}) = R_{0L}(u - w_L) \cdots R_{02}(u - w_2)R_{01}(u - w_1), \tag{4}$$

where $\mathbf{w} = (w_1, w_2, \ldots, w_L)$ is a vector of $w_i$, free inhomogeneity parameters. $T(u, \mathbf{w})$ is called the *monodromy matrix* and can be used to construct the transfer matrix $t(u, \mathbf{w})$ as

$$t(u, \mathbf{w}) = \text{tr}_0 \, T_0(u, \mathbf{w}), \tag{5}$$

where $\text{tr}_0$ identifies the partial trace over the auxiliary space (0).

By using the RTT relation (3), it follows that

$$[t(u, \mathbf{w}), t(v, \mathbf{w})] = 0. \tag{6}$$

$t(u, \mathbf{w})$ can be considered the generating function of the conserved charges $\mathbb{Q}_j$ of a quantum system. In fact, expanding the logarithm of $t(u, \mathbf{w})$ in a power series and calling $\mathbb{Q}_i$ the coefficient of $u^{i-1}$,

$$\mathbb{Q}_i = \partial_u^{i-1} \log t(u, \mathbf{w})|_{u=0}, \tag{7}$$

one has involutivity

$$[\mathbb{Q}_i, \mathbb{Q}_j] = 0, \quad i, j = 1, 2, \ldots \tag{8}$$

The involutivity condition holds for any values of the inhomogeneities $\mathbf{w}$. In many practical cases, $\mathbf{w} = \mathbf{0} = (0, 0, \ldots, 0)$ and all charges $\mathbb{Q}_i$ are translationally invariant sums of local operators of interaction range $i$, if the R-matrix obey the regularity condition $R(0) = P$ where $P$ is a permutation (see below). For values of $w_i$ close to 0, the charges are typically quasi-local, [34].

Usually, when considering a physical model, we are interested in its dynamics. For the continuous time evolution, the operator that generates the dynamics is the Hamiltonian $\mathbb{Q}_2 = H$, connected to the transfer matrix via[2]

$$H = \partial_u \log t(u, \mathbf{0})|_{u=0}. \tag{9}$$

In this work, we first focus in the discrete time evolution. The $R$-matrix of an integrable model can be used as the quantum gate to build an integrable quantum circuit. In section 2.2, we review the trotterization procedure [8, 9]: the inhomogeneities can be fixed to define a two-steps Floquet dynamics that preserves integrability for both periodic and open boundary conditions.

---

[1]The classical YBE takes a different expression than (2), we refer to [32, 33] for a detailed explanation.

[2]The log guarantees the locality of the charges for periodic boundary condition. For open boundary condition, (9) is $H = \partial_u t(u, \mathbf{0})|_{u=0}$ for a spin chain with open boundary condition.

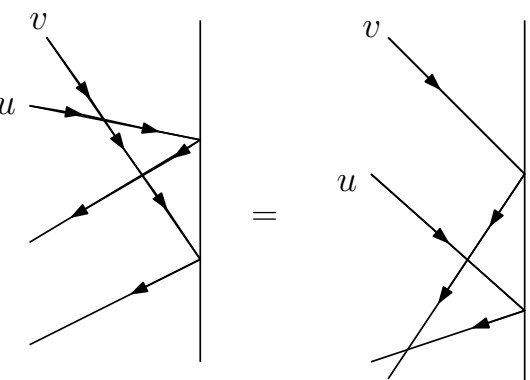

Figure 1: Graphical representation of the right reflection algebra (10).

### 2.1.2  Open boundary condition

For an integrable model with open boundary conditions, other than the usual Yang-Baxter equation (2), we have to introduce two new matrices $K^R(u)$ and $K^L(u)$ that take into account the right and left boundary of the spin chain. They are called *reflection matrices* or, in the same spirit of Eqs. (3), reflection algebra.

To avoid confusion, we indicate the $K$-matrices as $K_a^b$, where $a$ indicates the site where the matrix is acting non-trivially, while the label $b \in \{L, R\}$ denotes the left ($L$) or right ($R$) boundary matrix. Extra superscript may indicate the inverse of the matrix ($-1$) or the transpose ($t$) or the partial transpose ($t_1$) and ($t_2$) with respect to the first or to the second space.

$K^R$ describes the scattering of a particle moving towards the right wall and it satisfies the right boundary Yang-Baxter equation

$$R_{12}(u-v)K_1^R(u)R_{21}(u+v)K_2^R(v) = K_2^R(v)R_{12}(u+v)K_1^R(u)R_{21}(u-v). \tag{10}$$

Graphically, this corresponds to Fig. 1 (by reading the process from top to bottom).

To write the equation for $K^L$, we first[3] require that the $R$-matrix satisfies the following properties

a. Regularity $R(0) = P$, with $P$ the permutation operator in $V \otimes V$, acting as $P|a\rangle \otimes |b\rangle = |b\rangle \otimes |a\rangle$.

b. Symmetricity $R_{12}(u) = R_{21}(u)$ and $R_{12}^{t_1} = R_{12}^{t_2}$.

c. Unitarity $R_{12}(u)R_{12}(-u) = \rho(u)$.

d. Crossing Unitarity $R_{12}^{t_1}(u)R_{12}^{t_1}(-u-2\eta) = \tilde{\rho}(u)$,

where $\rho(u)$ and $\tilde{\rho}(u)$ are scalar functions, $\eta$ is a parameter that depends on the $R$-matrix of the model and $t_1$ and $t_2$ are the partial transpositions in the first and second space, respectively.

Under these conditions, $K^L$ satisfies the dual reflection equation

$$R_{12}(-u+v)K_1^{L\,t_1}(u)R_{21}(-u-v-2\eta)K_2^{L\,t_2}(v) = K_2^{L\,t_2}(v)R_{21}(-u-v-2\eta)K_1^{L\,t_1}(u)R_{12}(-u+v). \tag{11}$$

In Appendix C, we write down the transformation that can be performed on the $K^{R/L}$ matrices such that they remain solutions of (10)/(11).

---

[3]At the end of this section, we relax the crossing unitarity property d.

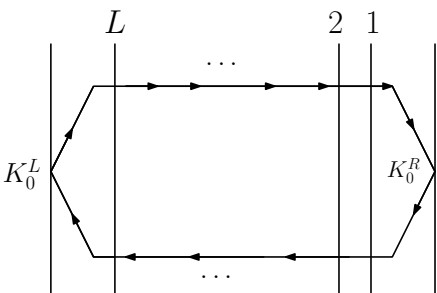

Figure 2: Double row transfer matrix (12).

The solutions of (10) and (11) are related by one of the following automorphisms

- $K^L(u) = K^{R\,t}(-u-\eta)$,

- $K^L(u) = \left((K^R)^{-1}\right)^t(u+\eta)$,

- $K_1^L(u) = \mathrm{tr}_2\Big(P_{12}R_{12}(-2u-2\eta)K_2^R(u)\Big)$.

It can be shown that the three automorphism are invertible and that by plugging the mapping for $K^R/K^L$ into (10)/(11), one obtains (11)/(10). In this way, we can conclude that all the solutions of $K^L$ are related to the ones for $K^R$.

By following Sklyanin's construction [3], the generator of the conserved charges is the *double row transfer matrix*

$$t(u) = \mathrm{tr}_0\Big(K_0^L(u)T(u)K_0^R(u)\hat{T}(u)\Big),\tag{12}$$

where

$$T(u) = R_{0L}(u)R_{0,L-1}(u)\cdots R_{01}(u),\tag{13}$$
$$\hat{T}(u) = R_{10}(u)R_{20}(u)\cdots R_{L0}(u).\tag{14}$$

We have used the unitarity property c. of the *R*-matrix. If this property does not hold, $\hat{T}(u)$ should be substituted by $T^{-1}(-u)$.

Graphically, we can think about the process as depicted in Fig. 2, [35]. This describes an auxiliary particle scattering through all the others, hitting the left wall, reflecting with opposite momenta, and after scattering all the others again, hitting the right wall.[4]

By using the Yang-Baxter equation (2) and the reflection equations (10) and (11), one can prove that, [3,36]

$$[t(u),t(v)] = 0.\tag{15}$$

More generally, we can define the inhomogeneous transfer matrix $t(u,\mathbf{w})$

$$t(u,\mathbf{w}) = \mathrm{tr}_0\Big(K_0^L(u)T(u,\mathbf{w})K_0^R(u)\hat{T}(u,\mathbf{w})\Big),\tag{16}$$

with

$$T(u,\mathbf{w}) = R_{0L}(u-w_L)R_{0,L-1}(u-w_{L-1})\cdots R_{01}(u-w_1),\tag{17}$$
$$\hat{T}(u,\mathbf{w}) = R_{10}(u+w_1)R_{20}(u+w_2)\cdots R_{L0}(u+w_L).\tag{18}$$

---

[4]We remark that, to simplify the notation, we are referring to $K^L(u)$ as the solution of the dual reflection equation (11). This is the object entering in the Fig. 2. If, on the other hand, we were to draw the equivalent of Fig. 1 for the left boundary, we would obtain an equation for the left reflection algebra ($\tilde{K}^L(u)$). The two matrices are related by a transformation as shown in [8], specifically, $K_1^L(u) = \mathrm{tr}_0\Big(\tilde{K}_0^L(-u)R_{01}(-2u)P_{01}\Big)$.

Similar to before, it can be shown that

$$[t(u, \mathbf{w}), t(v, \mathbf{w})] = 0 \,. \tag{19}$$

As for periodic boundary condition, for the continuous time dynamics, the derivative $\partial_u t(u, \mathbf{0})|_{u=0}$ defines a quantum spin chain Hamiltonian. In the next section, strictly following [8], we show how to fix the values of the inhomogeneities to build a discrete time process.

We remark that, the boundary equation for $K^L$ (11) holds if the $R$-matrix satisfies the crossing unitarity property $R^{t_1}(u)R^{t_1}(-u - 2\eta) \propto \mathbb{I}$. This can be understood by following the proof by Sklyanin, in [3].

In the absence of this property, one can follow the same proof and only use the unitarity property. In this way, the dual reflection equation for $K^L$ is [8,37]

$$R_{12}(-u + v)K_1^{L\,t_1}(u)\left(\left(R_{21}^{t_2}\right)^{-1}\right)^{t_2}(u + v)K_2^{L\,t_2}(v) = K_2^{L\,t_2}(v)\left(\left(R_{21}^{t_2}\right)^{-1}\right)^{t_2}(u + v)K_1^{L\,t_1}(u)R_{12}(-u + v) \,. \tag{20}$$

The automorphism that connects $K^L$ with $K^R$ for models without crossing unitarity is[5]

$$K_1^L(u) = \mathrm{tr}_2 P_{12}\left(\left(R_{12}^{t_2}\right)^{-1}\right)^{t_2}(2u)K_2^R(u) \,. \tag{21}$$

## 2.2 Two-step Floquet dynamics

Here we show how to fix the values of the inhomogeneity parameters $\mathbf{w}$ of the transfer matrix for both the periodic and open boundary conditions, in order to define an integrable discrete time dynamics. The original idea comes from the work by Baxter [7] where the value of the inhomogeneities are fixed to define a discrete time parallel update on the periodic lattice. Our construction initially follows [8,9]. In this work, our main focus is the case of open boundary conditions. For the reasons that will be clear in the following, we construct a quantum circuit where each gate is composed by the tensor product of two $R$-matrices of a spin $1/2$ chain. Due to the tensor product structure of local Hilbert space, it is easy to see that a (trivial) solution of the boundary reflection algebra can be found by taking the tensor product of two solutions of the reflection algebra for the single $R$-matrix, see Appendix A. We explore the non-trivial cases where the solutions of the boundary reflection algebra in the enlarged Hilbert space is richer.

For completeness, we first review the construction of the integrable circuit for the cases of periodic and open boundary conditions, in section 2.2.1 and 2.2.2. For completeness, in Appendix B, we also consider twisted boundary conditions.

### 2.2.1 Periodic boundary condition

For the case of periodic boundary conditions, the dimension of the lattice $L$ should be even. The inhomogeneities need to be fixed to

$$w_{2n} = \kappa \,, \qquad w_{2n+1} = -\kappa \,. \tag{22}$$

The transfer matrix (5) is

$$t(u) = \mathrm{tr}_0\Big(R_{0L}(u - \kappa)R_{0,L-1}(u + \kappa)\cdots R_{02}(u - \kappa)R_{01}(u + \kappa)\Big) \,. \tag{23}$$

We define the building block of the dynamics, the local 2-qubit gate[6]

$$U_{i,j} = \check{R}_{i,j}(2\kappa) = P_{i,j}R_{i,j}(2\kappa) \,. \tag{24}$$

---

[5]We thank R. Nepomechie for pointing out this automorphism between the two algebras.

[6]For the models we considered, this operator is unitary for appropriate normalization and imaginary $\kappa$.

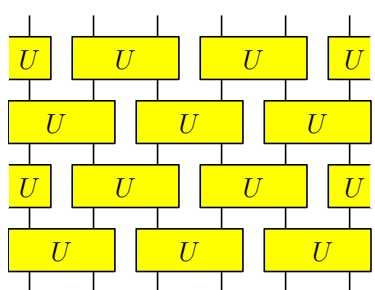

Figure 3: Quantum circuit with periodic boundary conditions.

At the points $u = \pm\kappa$, by using the properties of the $R$-matrix given at the beginning of section 2.1.2, $P_{i,j}A_iP_{i,j} = A_j$ and $\mathrm{tr}_0 P_{0,i} = \mathbb{I}$, the transfer matrix is

$$t(\kappa) = P_{1,L}P_{1,L-1}\cdots P_{1,3}P_{1,2}U_{2,3}\cdots U_{L-4,L-3}U_{L-2,L-1}U_{L,1}\,, \tag{25}$$

$$t(-\kappa) = P_{1,L}P_{1,L-1}\cdots P_{1,3}P_{1,2}\check{R}_{L-1,L}(-2\kappa)\check{R}_{L-3,L-2}(-2\kappa)\cdots\check{R}_{1,2}(-2\kappa)$$

$$= P_{1,L}P_{1,L-1}\cdots P_{1,3}P_{1,2}U_{L-1,L}^{-1}U_{L-3,L-2}^{-1}\cdots U_{1,2}^{-1}\,. \tag{26}$$

The dynamics is governed by the propagator $M$

$$M = t(-\kappa)^{-1}t(\kappa) = \mathbb{U}^o\mathbb{U}^e\,, \tag{27}$$

where now

$$\mathbb{U}^o = \prod_{k=1}^{L/2}U_{2k-1,2k}\,, \qquad \mathbb{U}^e = \prod_{k=1}^{L/2}U_{2k,2k+1}\,. \tag{28}$$

The operator $M$ describes one period of Floquet dynamics composed of two steps: $\mathbb{U}^e$ and $\mathbb{U}^o$. $\mathbb{U}^e$ updates every pair of consecutive sites with the first gate starting to act from even positions, while $\mathbb{U}^o$ with the first gate starting to act from odd positions,[7] see Fig. 3.

This construction builds an integrable quantum circuit. In fact, since $[t(u,\mathbf{w}),t(v,\mathbf{w})] = 0$, fixing the inhomogeneities to (22), it is easy to see that

$$[M, t(z)] = 0\,, \tag{29}$$

with $t(z)$ given in (23).

### 2.2.2 Open boundary conditions

For the case with open boundary conditions, the length $L$ of the lattice should be odd. We fix the inhomogeneities to have the following values

$$w_1 = w_3 = w_5 = \cdots = w_L = \kappa\,, \qquad w_2 = w_4 = \cdots = w_{L-1} = -\kappa\,. \tag{30}$$

The transfer matrix (16) is now

$$t(u) = \mathrm{tr}_0\Big(K_0^L(u)R_{0L}(u-\kappa)\ldots R_{02}(u+\kappa)R_{01}(u-\kappa)K_0^R(u)R_{10}(u+\kappa)R_{20}(u-\kappa)\ldots R_{L0}(u+\kappa)\Big). \tag{31}$$

If we evaluate the transfer matrix for $u = \kappa$, we obtain

$$t(\kappa) = K_1^R(\kappa)U_{23}U_{45}\cdots U_{L-1,L}U_{12}U_{34}\cdots U_{L-2,L-1}\bar{K}_L^L(\kappa)\,, \tag{32}$$

---

[7]We use the convention to enumerate the sites from right to left.

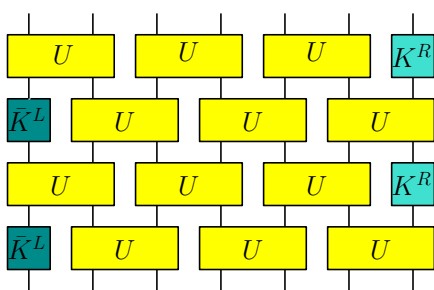

Figure 4: Quantum circuit with open boundary conditions.

where $\bar{K}^L(\kappa)$ is related to $K^L$ via

$$\bar{K}_L^L(\kappa) = \mathrm{tr}_0\big(K_0^L(\kappa)R_{0L}(2\kappa)P_{0L}\big).\tag{33}$$

In this case, we can define

$$M = t(\kappa) = \mathbb{U}^e\mathbb{U}^o,\tag{34}$$

where now

$$\mathbb{U}^o = \bar{K}_L^L(\kappa)\left(\prod_{k=1}^{\frac{L-1}{2}}U_{2k-1,2k}\right),\qquad \mathbb{U}^e = \left(\prod_{k=1}^{\frac{L-1}{2}}U_{2k,2k+1}\right)K_1^R(\kappa).\tag{35}$$

Similar to before, the operators $\mathbb{U}^e$ and $\mathbb{U}^o$ do not commute: $\mathbb{U}^e$ is responsible for the update of pair of consecutive sites, with the first one located at even position, while $\mathbb{U}^o$ with the first sites located at odd positions. They can be thought as a two-steps Floquet dynamics.

The graphical representation of this circuit is given in Fig. 4.

The circuit constructed in this way is integrable since the generator of the dynamics commutes with the transfer matrix for any value of the spectral parameter

$$[M, t(z)] = 0,\tag{36}$$

with $t(z)$ given in (31).

## 3 Our construction

In this section, we first discuss in 3.1, the construction of the quantum circuit we are going to study and we clarify the main question we would like to address. In 3.2, we discuss in details the steps that we performed.

### 3.1 Building the quantum circuit

In this section, we build up the circuit we are interested into. We first consider an arbitrary integrable spin $1/2$ chain characterized by the $R$-matrix $r(u)$ solution of the YBE in $\mathrm{End}(\mathbb{C}^2 \otimes \mathbb{C}^2 \otimes \mathbb{C}^2)$. We refer to $k^R(u), k^L(u) \in \mathrm{End}(\mathbb{C}^2)$ as the solutions of the right and left reflection algebra corresponding to the model, respectively.

For the reason that we briefly mentioned in the introduction and that will be clarified in the following, we construct a quantum circuits where each gate is composed by the tensor product of two copies of the $r(u)$ matrix, $R(u) = r(u) \otimes r(-u)$.

We refer to lower case letters for quantities on the initial spin $1/2$ chain and capital letters for quantities in this enlarged (doubled) Hilbert space.

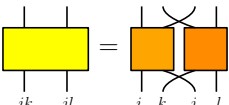

Figure 5: Elementary quantum gate given as the tensor product of two qubit.

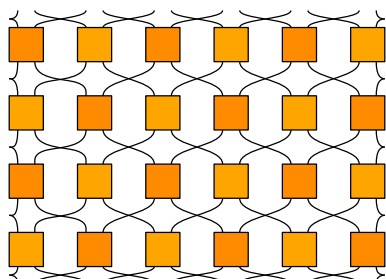

Figure 6: Periodic quantum circuit with factorized gate in the bulk.

It is easy to see that if $r(u)$ satisfies the YBE in $\text{End}\big(\mathbb{C}^2 \otimes \mathbb{C}^2 \otimes \mathbb{C}^2\big)$, $R(u)$ satisfies the YBE in $\text{End}\big(W \otimes W \otimes W\big)$, where $W = \mathbb{C}^2 \otimes \mathbb{C}^2$. The main question that we address in this paper is what happens to the boundary reflection algebra in this enlarged spin chain. As already mentioned, it is clear that, given $k^R$ and $k^L$, one can construct $K^R$ and $K^L$ just by taking their tensor product (Appendix A). However, as we describe in the following section, for a class of models, the solutions of the boundary reflection algebra in the enlarged Hilbert space can be much richer.

To keep track of the initial Hilbert space $\mathbb{C}^2$, we label the indices of the $R$-matrix indicating the spaces where it is acting non-trivially

$$U_{ik,jl} = \check{R}_{ik,jl}(u) = \check{r}_{ij}(u)\check{r}_{kl}(-u). \tag{37}$$

We represent this "double gate" as in Fig. 5, [39].

We can easily check that if the circuit built from $r$ is integrable, then the one with $R$ is also integrable and it is represented in Fig. 6.

For open boundary condition, a natural generalization of the right and left reflection algebra in the enlarged space is to consider $k^{R/L}(u)$ and $\tilde{k}^{R/L}(u)$ as two solutions of the boundary Yang-Baxter equation (10) and (11) for $r(u)$ and construct the solution of the boundary Yang-Baxter equation for $R(u)$ as $K_{ij}^{R/L}(u) = k_i^{R/L}(u)\tilde{k}_j^{R/L}(-u)$, see Appendix A for a proof of this statement. The two-steps Floquet circuit for this construction is Figure 7.

At this point, the main question of this work becomes clear: is it possible, that for some integrable models characterized by $r(u)$, there are some extra solutions of the boundary Yang-Baxter equation that cannot be written as $k \otimes \tilde{k}$? This corresponds to Fig. 8, where now the boundary gates obey the relations of Fig. 9.

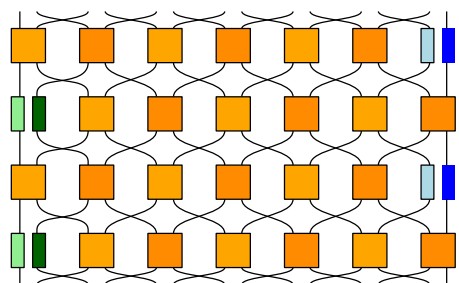

Figure 7: Open quantum circuit with factorizable boundaries.

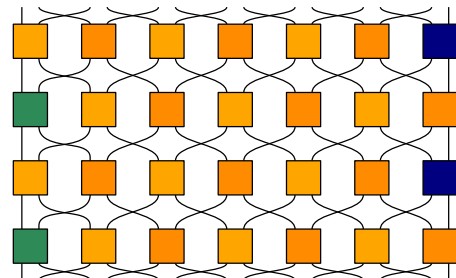

Figure 8: Open quantum circuit with both left and right boundaries non-factorizable.

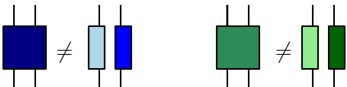

Figure 9: Conditions for non-factorized gates.

The answer to this question is positive. In the next section, we analyze for which models it holds.

## 3.2 Key steps

In this section, we first discuss the steps we performed in our constructions. We consider different integrable models of spins $1/2$ and for each of them we explore if there exist non-factorizable solutions of the boundary Yang-Baxter equation. We consider separately the cases: "bulk-interacting models" where we consider the XXX and the XXZ spin chain and "bulk-non-interacting models" where we consider the XX spin chain. For the last case, we analyze the XX spin chain in presence of magnetic field with either arbitrary strength $h$ or $h = 1$.

The steps we performed for each cases are

- Pick an integrable model characterized by $r(u)$.

- Built the gate of the quantum circuit as $U_{ij,kl} = \check{R}_{ij,kl}(2\kappa) = \check{r}_{ik}(2\kappa)\check{r}_{jl}(-2\kappa)$.

- Solve the reflection equation (10) for $K^R(u)$.

- Solve the reflection equation[8] (11) for $K^L(u)$.

- Build the integrable quantum circuit with open boundary condition by following the procedure of section 2.2.2.

To solve the equation (10), we consider the most general form for the $K^R$-matrix, as a $4 \times 4$ matrix with entries being functions dependent on a spectral parameter $u$. We label the entries of the $K^R$ matrix $\mathcal{K}_{i,j}(u)$, with $i,j$ identifying the row or the column where this element is placed. The expression of the $R$-matrix depends on the model we are analyzing. We normalize[9] the element $\mathcal{K}_{2,2}$ to one. A possible strategy to solve the boundary Yang-Baxter equation is to use a differential approach called Abel's method. For a review of the method,

---

[8]This solution can be found by applying to the $K^R$ solution the automorphism (21). If the $r(u)$ of the model analyzed obeys the crossing unitarity property, this automorphism reduce to the three (equivalent) given in section 2.1.2.

[9]Two $K$ matrices that differ only by a normalization factor solve the same boundary YBE. We choose $\mathcal{K}_{2,2} = 1$ since we require that $K^R(0) = \mathbb{I}$. Choosing to normalize an off-diagonal element to one won't be compatible with the regularity condition.

see [40]. This method consists of taking the formal derivatives of the equation (10) with respect to the spectral parameters $u$, then evaluate it at zero and use the regularity condition

$$K^R(0) = \mathbb{I}, \qquad \mathcal{K}_{i,j}(0) = \delta_{i,j}. \tag{38}$$

We did not impose any boundary condition on the first derivative of $K^R$, but for simplicity we indicate $\mathcal{K}'_{i,j}(0) = \kappa_{i,j}$.

We take the formal derivative of the boundary Yang-Baxter equation (10),

$$\partial_u(10)|_{u\to 0} = 0. \tag{39}$$

In this way, we obtain a system of equations depending on the variables $\mathcal{K}_{i,j}(v)$ and $\kappa_{i,j}$. The advantage to solve this system, compared to the initial one given by (10), is that now the equations are linear in the variables. To find all the solutions, we treat $\mathcal{K}_{i,j}(v)$ as independent from $\kappa_{i,j}$.[10] We solved some of the equations for some of the variables and plugging back into the remaining equations we solved for the others. Last, we fixed the remaining unknown functions by imposing the compatibility conditions that $\mathcal{K}_{i,j}{}'(0) = \kappa_{i,j}$. After having found all the solutions, we discard the ones leading to incompatible conditions.[11] We also discard the solutions that are not compatible with the regularity conditions.

This method is general and only requires that the functions $\mathcal{K}_{i,j}(v)$ are differentiable. This condition, for our case, is not restrictive at all since, as explained in section 5, the $K^R$ is one of the building block of the transfer matrix and hence, its derivatives correspond to the conserved charge characterizing the integrable model. Furthermore, (10) contains both the $u$ and $v$, spectral parameter dependence, taking the derivative w.r.t. $u$ and later sending it to zero, won't imply that we are finding solutions only at order linear in $u$.

This method was vastly used in the literature, in particular Lima-Santos and collaborators, classified the $K$-matrices associated with non-exceptional Lie algebras and superalgebras, see for example [41, 42].

## 4 Results

In this section, we discuss our result. We divide the section depending on the type of integrable models we considered in the bulk: interacting (section 4.1) or non interacting (4.2). At the end of this section, in 4.3 we discuss for which values of spectral parameter and free parameters, the quantum gates obtained are unitary.

### 4.1 Bulk-interacting models

We have analyzed the isotropic (XXX) and anisotropic (XXZ) Heisenberg spin chains.

#### 4.1.1 Heisenberg (XXX) spin chain

We consider the $r$-matrix of the Heisenberg spin chain[12]

$$r_{ij}(u) = p_{ij} + u\,\mathbb{I}, \tag{40}$$

---

[10]The choice on how many equations to solve in the first place depends on the model considered. So far, we did not find a systematic way to make this choice.

[11]An example of incompatible conditions may be that the element $\mathcal{K}_{2,2}(v) = 3$ and $\kappa_{2,2} = 5$.

[12]We remark that we are free to renormalize the $r$-matrix. If we want to find the continuous time dynamics generated by the Hamiltonian, we have to choose the normalization such that $p\partial_u r(u)|_{u=0}$ is Hermitian. To preserve unitarity, we have to choose the spectral parameter to be $u = i\tau$, with $\tau \in \mathbb{R}$.

where $\mathbb{I}$ is the identity operator and $p_{ij}$ is the permutation operator in $\mathbb{C}^2 \otimes \mathbb{C}^2$. We solve the boundary Yang-Baxter equation (10) for $K^R(u)$ and

$$R_{ij,kl}(u) = (p_{ik} + u\mathbb{I})(p_{jl} - u\mathbb{I}). \tag{41}$$

We solved the equation (10), as explained in the previous section, by using the Mathematica software and we obtained all the solutions for the right reflection matrix $K^R(u)$. After detailed investigation, we found that *all* the solutions of the boundary YBE for two copies of the Heisenberg spin chain take the factorized form

$$K_{ij}^R(u) = k_i^R(u)\tilde{k}_j^R(-u). \tag{42}$$

Both $k_i(u)$ and $\tilde{k}_j(u)$ are solutions[13] of the boundary YBE (10) for $r(u)$.

The Heisenberg spin chain is widely studied, diagonal solutions were classified in [3] and general solutions in [43]. For completeness, we write here[14] the most general solution of the reflection algebra for the XXX spin chain with $k^R$ regular,

$$k^R(u) = \begin{pmatrix} \frac{\kappa_1 u+1}{1-\kappa_1 u} & \frac{\kappa_2 u}{1-\kappa_1 u} \\ \frac{\kappa_4 u}{1-\kappa_1 u} & 1 \end{pmatrix}, \tag{43}$$

where $\kappa_1$, $\kappa_2$ and $\kappa_4$ are free constants. To construct the most general $K^R(u)$ from (42), one can assume that $\tilde{k}_j(u)$ has the same form as (43) and relabel the constants $\kappa_i$ as $\tilde{\kappa}_i$ since now they can in principle take different values. This is described in more details in the Appendix A.

To obtain the solution of the left boundary $K^L$ one can use the automorphism (21). This guarantees that if the solution for $K^R$ can be written in the factorized form, then the one for $K^L$ also factorizes. The $R$-matrix $R(u)$ (41) lacks the crossing unitarity property, thus the solution for $K^L$ can be obtained by employing the automorphism (21). The reason is that, for the Heisenberg spin chain, the crossing parameter for $r(u)$ is $\eta = 1$, so the crossing unitarity property is broken for $R(u) = r(u) \otimes r(-u)$. However, since we know from the more general automorphism that the solutions for $K^L$ should also factorize, we can separately obtain the solution for $k^L(u)$ and $\tilde{k}^L(u)$. In this way, we obtain

$$K^L(u) = (k^R)^t(-u-1) \otimes (\tilde{k}^R)^t(u-1). \tag{44}$$

The two-step Floquet circuit is the one in Figure 7.

### 4.1.2 XXZ spin chain

For the case of the XXZ spin chain, we consider the $r$-matrix[15]

$$r(u) = \begin{pmatrix} 1 & 0 & 0 & 0 \\ 0 & \sin(u)\csc(u-i\gamma) & -i\sinh(\gamma)\csc(u-i\gamma) & 0 \\ 0 & -i\sinh(\gamma)\csc(u-i\gamma) & \sin(u)\csc(u-i\gamma) & 0 \\ 0 & 0 & 0 & 1 \end{pmatrix}, \tag{45}$$

where $\gamma$ is related to the anisotropy $\Delta$ of the XXZ model by $\Delta = \cosh\gamma$.

After constructing the quantum gate $R(u)$ and repeating the steps explained in the previous section, we again obtain that *all* the solutions of the right Yang-Baxter equation are

$$K_{i,j}^R(u) = k_i^R(u)\tilde{k}_j^R(-u), \tag{46}$$

---

[13]We identify as $\tilde{k}$ the $K$-matrix acting on the site $j$ since this solution can be independent from $k(u)$. See Appendix A for details.

[14]This solution is related to the one of [43] by a reparametrization.

[15]As mentioned in the footnote 12, to guarantee that the dynamics is generated by an Hermitian operator, the normalization of the $r$ matrix should be $i\,r(u)$.

where $k_i$ is

$$k(u) = \begin{pmatrix} \frac{\kappa_{1,1}\sin u + 2\cos u}{2\cos u - \kappa_{1,1}\sin u} & \frac{2\kappa_{1,2}}{2\csc u - \kappa_{1,1}\sec u} \\ \frac{2\kappa_{2,1}}{2\csc u - \kappa_{1,1}\sec u} & 1 \end{pmatrix}, \tag{47}$$

and $\tilde{k}(u)$ is the same by substituting $\kappa$ with $\tilde{\kappa}$. This solution reproduces (43), in the same spirit as one gets the rational $R$-matrix of the Heisenberg spin chain from the trigonometric solution for the XXZ spin chain, [33]. As observed in [43], the solution of the right boundary Yang Baxter equation is independent from the parameter $\gamma$ (or equivalently $\Delta$). However, from the automorphisms (21) it is clear that $K^R$ is connected to $K^L$ via the $R$-matrix and hence the anisotropy $\Delta$ (or $\gamma$) of the spin chain enters. We can repeat the same argument of the Heisenberg spin chain, and we obtain

$$K^L(u) = (k^R)^t(-u + i\gamma) \otimes (\tilde{k}^R)^t(u + i\gamma). \tag{48}$$

We conclude that, also for the case of the XXZ spin chain, all the solutions of the boundary Yang Baxter equation factorize and the quantum integrable circuit built from this solution is the one in Figure 7.

## 4.2 Bulk-non-interacting models

Now, we consider the XX spin chain and we build the quantum circuits starting from this elementary gate. Afterward, we add the contribution of the magnetic field along the $z$-direction.

### 4.2.1 The XX spin chain

The $r$-matrix of the XX spin chain is

$$r(u) = \begin{pmatrix} 1 & 0 & 0 & 0 \\ 0 & \tan(u) & \sec(u) & 0 \\ 0 & \sec(u) & \tan(u) & 0 \\ 0 & 0 & 0 & 1 \end{pmatrix}. \tag{49}$$

After constructing the quantum gate $R(u)$, differently from the previous two cases, by solving the right boundary Yang-Baxter equation (10), we obtain two independent types of solutions

1. $K^{R,(f)}_{ij}(u) = k_i(u)\tilde{k}_j(-u)$,

2. $K^{R,(nf)}_{ij}(u) \neq k_i(u)\tilde{k}_j(-u)$.

The subscript (f) or (nf) are used to indicate whether the $K$ matrix is factorized (f) or not (nf).

For the solution of the type f, as for the interacting case, $k_i$ and $\tilde{k}_j$ are solutions of the one layer spin chain and take the form (47).

The solution of type nf are the most interesting and will allow us to answer positively to the question posed at the beginning. In fact, for this case, considering the boundary Yang-Baxter equation in a larger space where the $R$-matrix is the tensor product of two $r$-matrices, allows for a non-factorized form of the solutions. We solved the boundary YBE (10) as explained in sec. 3.2 by starting from the most general Ansatz for the $K^R$-matrix with all the 16 elements and we obtained that all the solutions of the right boundary Yang-Baxter equation are of 8 vertex type, explicitly

$$K^R(u) = \begin{pmatrix} \frac{\kappa_{1,1}-\kappa_{3,3}+2\cot u}{q(u)}-1 & 0 & 0 & \frac{\kappa_{1,4}}{q(u)} \\ 0 & 1 & \frac{\kappa_{2,3}}{q(u)} & 0 \\ 0 & \frac{\kappa_{3,2}}{q(u)} & \frac{\kappa_{3,3}}{q(u)}+1 & 0 \\ \frac{\kappa_{4,1}}{q(u)} & 0 & 0 & \frac{2\cot u-\kappa_{1,1}}{q(u)}-1 \end{pmatrix}, \tag{50}$$

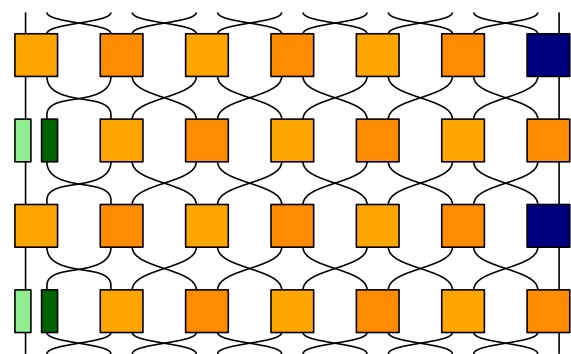

Figure 10: Circuit built from the solutions of the boundary YBE $K^{L,(\mathrm{f})}$ and $K^{R,(\mathrm{nf})}$.

where

$$q(u) = \frac{1}{4\cot u}\left(\kappa_{2,3}\kappa_{3,2} + \kappa_{1,1}\left(\kappa_{3,3} - \kappa_{1,1}\right) - \kappa_{1,4}\kappa_{4,1} - 2\kappa_{3,3}\cot u + 4\cot^2 u\right). \tag{51}$$

There are 6 free parameters: $\kappa_{1,1}, \kappa_{3,3}, \kappa_{2,3}, \kappa_{3,2}, \kappa_{1,4}, \kappa_{4,1}$.

It is clear that this $K^R(u)$ matrix cannot be written as the tensor product of two $2 \times 2$ $k$-matrices, hence it is not factorized.

To obtain the solution of the $K^L(u)$, we use one of the three (equivalent) automorphisms of [3], summarized in section 2.1.2. This model has crossing unitarity symmetry[16] and $\eta = \pi/2$.

We start from one of the two different solutions found for $K^R$ and we can construct the corresponding $K^L$, explicitly

$$K^{L,(i)}(u) = \left(K^{R,(i)}\right)^t(-u - \pi/2), \quad i \in \{\mathrm{f}, \mathrm{nf}\}. \tag{52}$$

For this model, we are allowed to build 4 different type of transfer matrices. In fact, there are two different families of $K^R$ matrices that solve the right boundary Yang-Baxter eq. (10), which we labelled $K^{R,(\mathrm{f})}$ and $K^{R,(\mathrm{nf})}$ and two that solve the eq. (11) ($K^{L,(\mathrm{f})}$, $K^{L,(\mathrm{nf})}$).

By following Sklyanin's proof [3] of the commutativity between $[t(u), t(v)] = 0$ for (16), one can similarly prove that,

$$t^{(i,j)}(u, \mathbf{w}) = \mathrm{tr}_0\Big(K_0^{L,(i)}(u)T(u,\mathbf{w})K_0^{R,(j)}(u)\hat{T}(u,\mathbf{w})\Big), \tag{53}$$

$$[t^{(i,j)}(u,\mathbf{w}), t^{(i,j)}(v,\mathbf{w})] = 0, \quad \forall\, i, j \in \{\mathrm{f}, \mathrm{nf}\}. \tag{54}$$

The definition (53) for the transfer matrix $t(u)$ includes 4 transfer matrices. This type of construction was also considered in [44] for the Hubbard model.

The circuit built when $i = \mathrm{f}$ and $j = \mathrm{nf}$ can be represented as in Fig. 10.

For this type of circuit, the space-reflection symmetry is broken, while integrability is preserved. This type of circuit is equivalent to a finite free fermionic model with an interacting impurity. By fixing the parameters $\kappa_{1,4} = \kappa_{4,1} = 0$ in (50) and considering the unitarity conditions given in sec. 4.3, we obtain unitary dynamics with an interacting impurity with $U(1)$ symmetry. It would perhaps be interesting to analyze the transport properties of this model.

We note that since all the models obtained are Yang-Baxter integrable, they are explicitly solvable using the Bethe Ansatz or some other techniques from the theory of integrable models, for example the Baxter $Q$-operators. For the case of the algebraic Bethe Ansatz applied to quantum circuits with periodic boundary conditions, we refer to [45]. The algebraic Bethe Ansatz for open boundary conditions is analyzed in [3] for the case of a homogeneous transfer matrix

---

[16]Contrary to the interacting model, here the crossing unitarity property is preserved for the $R$-matrix in the enlarged space. The reason is that here $r(u - 2\eta) = r(u + 2\eta)$.

and diagonal boundaries. The inhomogeneous case is straightforward. The case with non-diagonal boundaries is more complex; however, numerous techniques have been developed to address it (see, for example, [46]). Furthermore, for the particular choice of parameters that corresponds to having a $U(1)$ symmetry, it is also possible to perform the coordinate Bethe Ansatz, in a manner similar to [47].

### 4.2.2 The XX spin chain in magnetic field

We consider the case of the XX spin chain in a magnetic field with strength $h$. This model is also referred to as the XXh model. The $r$-matrix is [48],

$$
r(u) = \begin{pmatrix} 1 - e^{i(u+2\psi)} & 0 & 0 & 0 \\ 0 & e^{i\psi}\left(1 - e^{iu}\right) & -2ie^{\frac{1}{2}i(u+2\psi)}\sin\psi & 0 \\ 0 & -2ie^{\frac{1}{2}i(u+2\psi)}\sin\psi & e^{i\psi}\left(1 - e^{iu}\right) & 0 \\ 0 & 0 & 0 & e^{iu} - e^{2i\psi} \end{pmatrix}, \tag{55}
$$

where $\cos\psi = h$ is the strength of the magnetic field.

Similar to the case treated in section 4.2.1, also in this case there are two type of solutions:

1. $K_{ij}^{R,(f)}(u) = k_i(u)\tilde{k}_j(-u)$,

2. $K_{ij}^{R,(nf)}(u) \neq k_i(u)\tilde{k}_j(-u)$.

Due to the presence of the magnetic field, the solutions of type f and nf have less free parameters as compared to the previous cases.

For the type f, only a diagonal solution is allowed,

$$
k^R(u) = \begin{pmatrix} \frac{2}{1 - \kappa_{1,1}\tan\left(\frac{u}{2}\right)} - 1 & 0 \\ 0 & 1 \end{pmatrix}. \tag{56}
$$

The solution of type nf is

$$
K^R(u) = \begin{pmatrix} \frac{\cot\left(\frac{u}{2}\right) - \kappa_{4,4}}{g(u)} - 1 & 0 & 0 & \frac{\kappa_{1,4}}{g(u)} \\ 0 & 1 & 0 & 0 \\ 0 & 0 & \frac{\kappa_{3,3}}{g(u)} + 1 & 0 \\ \frac{\kappa_{4,1}}{g(u)} & 0 & 0 & \frac{\kappa_{4,4} - \kappa_{3,3} + \cot\left(\frac{u}{2}\right)}{g(u)} - 1 \end{pmatrix}, \tag{57}
$$

where $g(u) = \frac{1}{2\cot\left(\frac{u}{2}\right)}\left(-\kappa_{4,4}^2 - \kappa_{1,4}\kappa_{4,1} + \kappa_{3,3}\left(\kappa_{4,4} - \cot\left(\frac{u}{2}\right)\right) + \cot^2\left(\frac{u}{2}\right)\right)$.

In this case, there are 4 free parameters: $\kappa_{1,4}, \kappa_{4,1}, \kappa_{4,4}, \kappa_{3,3}$. The presence of the magnetic field, similarly as for the factorized solution, is imposing some extra constraints on the solution of the boundary Yang-Baxter equation. This model has crossing symmetry, with crossing parameter $\eta = \pi$. We obtain the solution for $K^L$ by using one of the three (equivalent) automorphisms given in sec. 2.1.2,

$$
K^{L,(i)}(u) = \left(K^{R,(i)}\right)^t(-u-\pi), \quad i \in \{f, nf\}. \tag{58}
$$

Since also for the case with the magnetic field, we obtained two families of $K$-matrices, we can construct four different commuting transfer matrices as (53) and (54).

**XX spin chain in magnetic field with $h = 1$.** A particular limit of this model is when the strength of the magnetic field is $h = 1$. This case can be recovered from the previous one by properly renormalizing the $r$-matrix, rescaling the spectral parameter $u \to 4u\psi$ and then taking the limit $\psi \to 0$ ($h \to 1$). However, given the simplicity of the model, we write it separately. The $r$-matrix is polynomial

$$
r(u) = \begin{pmatrix} 1 & 0 & 0 & 0 \\ 0 & \frac{2u}{2u+1} & \frac{1}{2u+1} & 0 \\ 0 & \frac{1}{2u+1} & \frac{2u}{2u+1} & 0 \\ 0 & 0 & 0 & \frac{1-2u}{2u+1} \end{pmatrix}.
\tag{59}
$$

The factorizable solution of the boundary Yang-Baxter equation is

$$
K_{ij}^{R,(f)} = k_i^R(u)\tilde{k}_j^R(-u), \quad \text{where} \quad k^R(u) = \begin{pmatrix} \frac{2}{1-u\kappa_{1,1}} - 1 & 0 \\ 0 & 1 \end{pmatrix},
\tag{60}
$$

and the non-factorizable one is

$$
K^{R,(nf)}(u) = \begin{pmatrix} \frac{2/u - \kappa_{4,4}}{m(u)} - 1 & 0 & 0 & \frac{\kappa_{1,4}}{m(u)} \\ 0 & 1 & 0 & 0 \\ 0 & 0 & \frac{\kappa_{3,3}}{m(u)} + 1 & 0 \\ \frac{\kappa_{4,1}}{m(u)} & 0 & 0 & \frac{\kappa_{4,4} - \kappa_{3,3} - 2/u}{m(u)} - 1 \end{pmatrix},
\tag{61}
$$

with $m(u) = -\frac{1}{4u}(u^2\kappa_{4,4}^2 + u^2\kappa_{1,4}\kappa_{4,1} + u\kappa_{3,3}(2 - u\kappa_{4,4}) - 4)$.

For this model, the crossing unitarity property is broken even at the level of $r(u)$. We obtain the solutions $K^{L,(f)}$ and $K^{L,(nf)}$ from the automorphism (21),

$$
K_{i,j}^{L,(f)}(u) = k_i^L(u)\tilde{k}_j^L(-u), \quad \text{where} \quad k^L(u) = \begin{pmatrix} \frac{\kappa_1(2u-1)+1}{\kappa_1(2u+1)-1} & 0 \\ 0 & 1 \end{pmatrix},
\tag{62}
$$

$$
K^{L,(nf)}(u) = z(u) \begin{pmatrix} \frac{-8(\kappa_{3,3} - 2u\kappa_{4,4} + 2) + h(-2u) - h(-1)}{h(1-2u)} & 0 & 0 & \frac{16u\kappa_{1,4}}{h(1-2u)} \\ 0 & 1 & 0 & 0 \\ 0 & 0 & \frac{h(2u+1)}{h(1-2u)} & 0 \\ \frac{16u\kappa_{4,1}}{h(1-2u)} & 0 & 0 & \frac{-8(\kappa_{3,3} + 2u\kappa_{4,4} + 2) + h(2u) - h(-1)}{h(1-2u)} \end{pmatrix},
\tag{63}
$$

where $h(u) = u^2\kappa_{4,4}^2 + u^2\kappa_{1,4}\kappa_{4,1} - u\kappa_{3,3}(u\kappa_{4,4} + 4) - 16$ and $z(u) = \frac{(1-16u^2)h(1-2u)}{64u^2(6(u\kappa_{3,3}+2)+h(u))}$.

Also for this case, we can construct 4 different commuting transfer matrices as (53) and (54).

## 4.3 Unitarity of the $U$ and $K$ gates

In this paragraph, we discuss the unitarity property of the quantum gates that we used to build the quantum circuit. In particular, in section 2.2.2, we obtain that to build the quantum circuit for the case with open boundary condition, the main ingredients are the quantum gate $U$ given in (24) and $K^R$, solution of the boundary reflection algebra (10) and $\bar{K}^L$, connected to the solution of the reflection algebra by (33).

For simplicity, we introduce the notation $f_R$ and $f_I$. Given a parameter or a variables $f$, $f_R$ means $f \in \mathbb{R}$ and $f_I$ means $f = ig$, $g \in \mathbb{R}$.

The free constants appearing in the $K$ matrices are $\kappa_i$ or $\kappa_{i,j}$, depending on the model. Here, we refer to $\kappa_R$ or $\kappa_I$ as all the constants of the model under consideration.

We discuss the unitarity property separately for each of the model studied.

**XXX spin chain of sec. 4.1.1**

| | |
|---|---|
| gate $U$, | $u = u_I$, |
| gates $K^R, \bar{K}^L$, | $u = u_I$, $\kappa = \kappa_R$, $\kappa_2 = \kappa_4$, |
| | or $u = u_R$, $\kappa = \kappa_I$, $\kappa_2 = \kappa_4$. |

**XXZ spin chain of sec. 4.1.2**

| | |
|---|---|
| gate $U$, | $u = u_I$, $\gamma = \gamma_I$, |
| | or $u = u_R$, $\gamma = \gamma_R$, |
| gates $K^R, \bar{K}^L$, | $u = u_I$, $\kappa = \kappa_R$, $\kappa_{1,2} = \kappa_{2,1}$, |
| | or $u = u_R$, $\kappa = \kappa_I$, $\kappa_{1,2} = \kappa_{2,1}$. |

**XX spin chain of sec. 4.2.1**   No magnetic field

| | |
|---|---|
| gate $U$, | $u = u_I$, |
| gates $K^{R,(\mathrm{nf})}, \bar{K}^{L,(\mathrm{nf})}$, | $u = u_I$, $\kappa = \kappa_R$, $\kappa_{1,4} = \kappa_{4,1}$, $\kappa_{2,3} = \kappa_{3,2}$, |
| | or $u = u_R$, $\kappa = \kappa_I$, $\kappa_{1,4} = \kappa_{4,1}$, $\kappa_{2,3} = \kappa_{3,2}$, |
| gates $K^{R,(\mathrm{f})}, \bar{K}^{L,(\mathrm{f})}$, | $u = u_I$, $\kappa = \kappa_R$, $\kappa_{1,2} = \kappa_{2,1}$, |
| | or $u = u_R$, $\kappa = \kappa_I$, $\kappa_{1,2} = \kappa_{2,1}$. |

**XX spin chain of sec. 4.2.2**   Arbitrary strength of magnetic field $h$

| | |
|---|---|
| gate $U$, | $u = u_I$, $\psi = \psi_R$, |
| | or $u = u_R$, $\psi = \psi_I$, |
| gates $K^{R,(\mathrm{nf})}$, | $u = u_I$, $\kappa = \kappa_R$, $\kappa_{1,4} = \kappa_{4,1}$, |
| | or $u = u_R$, $\kappa = \kappa_I$, $\kappa_{1,4} = \kappa_{4,1}$, |
| gates $\bar{K}^{L,(\mathrm{nf})}$, | same as $K^{R,(\mathrm{nf})}$ and $\psi = \psi_R$ if $u = u_I$, |
| | or $\psi = \psi_I$ if $u = u_R$, |
| gates $K^{R,(\mathrm{f})}$, | $u = u_R$, $\kappa = \kappa_I$, |
| | or $u = u_I$, $\kappa = \kappa_R$, |
| gates $\bar{K}^{L,(\mathrm{f})}$, | same as $K^{R,(\mathrm{f})}$ and $\psi = \psi_R$ if $u = u_I$, |
| | or $\psi = \psi_I$ if $u = u_R$. |

**XX spin chain of sec. 4.2.2**   $h = 1$

In this case $h = \cos\psi = 1$, $\psi = 0$. The unitarity conditions are the same as for arbitrary $h$, without the condition on $\psi$.

# 5   Continuous time dynamics

In this section, we focus on the continuous time dynamics. In sec. 5.1, we obtain the continuous dynamics as a limit of the discrete one. We motivate our choice of the factorized form $R(u) = r(u) \otimes r(-u)$ of the quantum gate. In sec. 5.2, we discuss the connection between the boundary terms in the conserved charges and their Lindbladian structure.

## 5.1 Continuous time dynamics as the limit of discrete one

For Yang-Baxter integrable model, characterized by a regular $R$-matrix, it is easy to connect the continuous and discrete time dynamics. In fact, for these models, we can expand the $R$-matrix as

$$\check{R}_{i,j}(u) = \mathbb{I} + u\,\partial_u \check{R}_{i,j}(u)|_{u=0} + \ldots = \mathbb{I} + u\,q_{i,j}^{(2)} + \ldots, \tag{64}$$

with $q^{(2)}$ being the density of range 2 of one of the conserved charges of the integrable model under consideration.

We remind that we identified the quantum gate of the circuit as

$$U_{i,j} = \check{R}_{i,j}(2\kappa), \tag{65}$$

and the discrete dynamical evolution for two steps is generated by the $M$ operator given in (27)-(28) for the case with periodic boundary condition and (34)-(35) for the open one.

To obtain the continuous time evolution, [9], we set $2\kappa = -i\,\Delta t$ and $\Delta t = t/n$ and we apply $n$ times the two step discrete evolution $M$, with $n$ being large. In this way, we obtain the continuous time evolution

$$\lim_{n\to\infty} M^n = \exp(-i\,t\,\mathbb{Q}_2), \tag{66}$$

with $\mathbb{Q}_2 = \sum_i q_{i,i+1}^{(2)}$. The charge density $q^{(2)}$ is related to the $R$-matrix as

$$\partial_u \check{R}_{i,j}(u)|_{u\to 0} = q_{i,j}^{(2)}. \tag{67}$$

Until now, the construction was general and did not make use of our particular choice of the quantum gate.

We clarify the reason of our choice of the quantum gate $R(u) = r(u) \otimes r(-u)$ by adding the indices explicitly (37) and taking the limit $u \to 0$. The subscripts in the operator identify the space on the original spin 1/2 chain where the operators are acting non-trivially. The charge associated to this $R$-matrix is

$$\partial_u r_{ik}(u) r_{jl}(-u)|_{u\to 0} = q_{ij,kl}^{(2)} = h_{ik} - h_{jl} = h_{ik} - h_{jl}^t, \tag{68}$$

where in the last step we used the fact that the $r$-matrix is invariant under transposition.

At this stage, a step back is necessary. Initially, our motivation for exploring this specific problem was to determine whether the property of a model's steady state being integrable in a boundary-driven setup could also extend to the Yang-Baxter integrability of the spin chain with open boundary condition. Specifically, we considered the research from over a decade ago on constructing non-equilibrium steady state (NESS) density operators for boundary-driven, locally interacting quantum chains, where the driving is applied through Markovian dissipation channels (Lindblad evolution) at the chain's boundaries, [23–29].

In order to answer to this question, we started by considering the coherent time evolution of a density matrix $\rho$ via the Liouville-von Neumann equation,

$$\mathcal{L}[\rho] := i[H,\rho] = i\Big[\sum_i h_{i,i+1}, \rho\Big]. \tag{69}$$

We consider the dynamics governed by a local nearest-neighbor Hamiltonian $H = \sum_i h_{i,i+1}$ acting on the bulk of a spin 1/2 chain of $L$ sites. We can map this action in the "doubled" Hilbert space $\mathcal{H} \otimes \mathcal{H}^*$ by performing a vectorization

$$\text{End}(\mathcal{H}) \to \mathcal{H} \otimes \mathcal{H}^*. \tag{70}$$

In this super-space, the vectorized action of the superoperator $\mathcal{L}$ can be expressed as

$$\mathcal{L} = \sum_J \mathcal{L}_{J,J'} \equiv i \sum_{j=0} (h_{2j+1,2j+3} - h^t_{2j+2,2j+4}), \tag{71}$$

where with the sum over $J$ we are considering the neighbouring super-sites in the vectorized space $J = (2j+1, 2j+2)$, $J' = (2j+3, 2j+4)$. This coincides, up to a normalization factor, with the expression given in (68).

In the previous sections, by constructing the reflection algebra, we answered to the question whether we are allowed to add some non-factorizable terms to the boundary of the spin chain, such that the Yang-Baxter integrability of the model is preserved. We obtained that, if we consider an interacting model in the bulk of the spin chain, meaning the XXX or the XXZ spin chain, there are no interesting (non-factorizable) boundary term that preserves integrability. However, for the case of the XX spin chain, both with and without magnetic field, we obtained non-trivial solutions of the right (10) and left reflection algebra (11).

In [23, 24], it is shown that for the XXZ chain with Lindbladian spin source/sink on the left/right boundary, the NESS is proven to be exactly solvable in terms of a matrix product Ansatz. This, however, does not coincide with the full Yang-Baxter integrability of the spin chain. In fact, the only case where we found a non-factorizable solution is the case where in the bulk we have the XX model. We now focus on this type of solution to understand if it is possible to write the boundary terms in the Lindbladian form.

## 5.2 Boundary terms of Lindbladian form

We address this question by constructing the double row transfer matrix, as given in (12) and, by taking the higher order derivatives and evaluate it at $u = 0$, we calculate the analytical expression of the terms at the boundary. By taking the higher order derivative[17] of the double row transfer matrix, one can prove that the first non-trivial conserved charge is

$$\mathbb{Q}_2 = \sum q^{(2)}_{i,i+1} + q^{(2),L}_L + q^{(2),R}_1. \tag{72}$$

We would like to check, whether it is possible to find some local operators $\ell^{(i)}$ such that

$$q^{(2),L/R}_{L/1} = \sum_i \left[ \ell^{(i)}_{L/1} \otimes \ell^{(i)*}_{L/1} - \frac{1}{2}\left(\ell^{(i)\dagger}_{L/1} \ell^{(i)}_{L/1}\right) \otimes \mathbb{I} - \frac{1}{2}\mathbb{I} \otimes \left(\ell^{(i)t}_{L/1} \ell^{(i)*}_{L/1}\right) \right], \tag{73}$$

where the superscript $L/R$ of the $q^{(2)}$ identify respectively the left and right boundary and the subscript $L/1$ indicate in which site of the spin chain the operators are acting.

First, we have to obtain the analytical expression of the boundary terms $q^{(2),L/R}_{L/1}$ in terms of the quantities $K^{L/R}$ and the Hamiltonian $h$, or, more generally, the $R$-matrix.

### 5.2.1 Analytical expression of the boundary terms

For the class of models we are considering, except model of section 4.2.2 with $h = 1$, the first and second derivative of the transfer matrix do not generate the dynamics. In fact, the first derivative vanishes and the second derivative is proportional to the identity operator. The first non-trivial charge is the derivative of order 3 of the transfer matrix. For completeness, since we were not able to find this expressions in the literature, we give the analytical expression of the first two charges in the appendix D. We derived these expressions, as well as the one for the third derivatives, using Mathematica software and the non-commutative product. The precise

---

[17]We remark that for the models we are considering (except for the XX spin chain with magnetic field and $h = 1$), this charge corresponds to the third derivatives of $t(u)$.

expression for the third derivative in a general model is quite lengthy, so it is not included in this paper. However, upon request, we are happy to share the Mathematica notebook with anyone interested. For the model we are considering, many terms are zero and the final expression is very compact.

We obtained, by ignoring the term proportional to the identity matrix

$$\partial_u^3 t(u)|_{u\to 0} = \alpha\, q_{Bulk}^{(2)} + \frac{\alpha}{2}\,\partial_u K_1^R(0) + \mathrm{tr}_0\Big(2\,\partial_u K_0^L(0)\,h_{0L}^2 + \partial_u^2 K_0^L(0)\,h_{0L} + K_0^L(0)\,\partial_u^3 \check{R}_{0L}(0)\Big), \quad (74)$$

where

$$\alpha = \mathrm{tr}(\partial_u^2 K^L(0)) + 4\,\mathrm{tr}_0(K_0^L(0)\,h_{0L}^2) + 4\,\mathrm{tr}_0(\partial_u K_0^L(0)\,h_{0L})\,. \quad (75)$$

We can then consider the dynamics being generated by the charge

$$q^{(2)} = q_{Bulk}^{(2)} + q_L^{(2),L} + q_1^{(2),R}\,, \quad (76)$$

where

$$q_L^{(2),L} = \frac{1}{\alpha}\mathrm{tr}_0\Big(2\,\partial_u K_0^L(0)\,h_{0L}^2 + \partial_u^2 K_0^L(0)\,h_{0L} + K_0^L(0)\,\partial_u^3 \check{R}_{0L}(0)\Big)\,, \quad (77)$$

$$q_1^{(2),R} = \frac{1}{2}\partial_u K_1^R(0)\,. \quad (78)$$

### 5.2.2 Attempt to write the solution as Lindbladian

Having found the analytical expression, we can now check whether the boundary terms $q_L^{(2),L}$ and $q_1^{(2),R}$ can be brought into a Lindbladian form (73).

A direct approach consists in considering the most general local operator $\ell$ as

$$\ell^{(i)} = \begin{pmatrix} \lambda_{i,1} & \lambda_{i,2} \\ \lambda_{i,3} & \lambda_{i,4} \end{pmatrix}, \quad (79)$$

and find a solutions for the $\lambda_{i,j}$ such that the decomposition (73) matches (77)-(78) separately for the right and the left boundaries. However, this approach will be computationally very hard since the sum over $i$ in (73) runs over the square of dimension of the local Hilbert space.

Instead of solving the problem by following the direct approach, we found a shortcut.

**XX spin chain**  First, we calculated the boundary terms corresponding to the XX spin chain given in section 4.2.1 and to the non factorized solution (50) and we obtain

$$q_1^{(2),R} \propto \begin{pmatrix} \kappa_{1,1} & 0 & 0 & \kappa_{1,4} \\ 0 & 0 & \kappa_{2,3} & 0 \\ 0 & \kappa_{3,2} & \kappa_{3,3} & 0 \\ \kappa_{4,1} & 0 & 0 & \kappa_{3,3} - \kappa_{1,1} \end{pmatrix}, \quad (80)$$

and

$$q_L^{(2),L} \propto \begin{pmatrix} 2\tilde\kappa_{1,1} - \tilde\kappa_{3,3} & 0 & 0 & 2\tilde\kappa_{1,4} \\ 0 & -\tilde\kappa_{3,3} & 2\tilde\kappa_{2,3} & 0 \\ 0 & 2\tilde\kappa_{3,2} & \tilde\kappa_{3,3} & 0 \\ 2\tilde\kappa_{4,1} & 0 & 0 & \tilde\kappa_{3,3} - 2\tilde\kappa_{1,1} \end{pmatrix}. \quad (81)$$

If we call $k_1, k_2, k_3$ and $k_4$ the elements in the diagonal of one of these two matrices, they satisfy

$$k_1 + k_4 = k_2 + k_3\,. \quad (82)$$

First, we construct the matrix form of the operators $q_{L/1}^{(2),L/R}$ in (73), by using the definition (79) for the $\ell^{(i)}$. The condition (82) for the operator $\ell^{(i)}$ in (73) implies

$$\sum_i |\lambda_{i,1} - \lambda_{i,4}|^2 = 0, \tag{83}$$

which is the sum of positive quantities and it can be zero only if

$$\lambda_{i,1} = \lambda_{i,4}, \quad \forall\, i, \tag{84}$$

and since the matrices $\ell^{(i)}$ can be shifted by identity, we can set these quantities to zero.

This proves that the condition (82) could be satisfied only if $\ell^{(i)}$ has the structure

$$\ell^{(i)} = \begin{pmatrix} 0 & \lambda_{i,2} \\ \lambda_{i,3} & 0 \end{pmatrix} = \lambda_{i,2}\sigma^+ + \lambda_{i,3}\sigma^-. \tag{85}$$

At this point, to match the expression (73) is much easier and we obtain that, if we set[18]

$$\kappa_{3,2} = \kappa_{2,3}^*, \qquad \kappa_{1,1} = \frac{1}{2}(\kappa_{1,4} - \kappa_{4,1}), \qquad \kappa_{3,3} = 0, \tag{86}$$

and similar expressions for the $\tilde{\kappa}$, we can rescale the remaining free parameters such that the right and left boundaries are described by

$$\ell_1^{(1)} = \alpha\sigma^+, \qquad \ell_1^{(2)} = \beta_1\sigma^+ + \beta_2\sigma^-, \tag{87}$$

and similarly

$$\ell_L^{(1)} = \tilde{\alpha}\sigma^+, \qquad \ell_L^{(2)} = \tilde{\beta}_1\sigma^+ + \tilde{\beta}_2\sigma^-, \tag{88}$$

where $\alpha$, $\beta_1$ and $\beta_2$ (and their tilde versions) are free constants.[19]

We conclude that, the XX spin chain in the bulk, with boundary terms described by the jump operators (87) and (88) is Yang-Baxter integrable.

### 5.2.3 XX spin chain with magnetic field

We consider the XX spin chain with magnetic field given in section 4.2.2, we calculate the boundary term for the non factorized solution (57) and we obtain

$$q_1^{(2),R} \propto \begin{pmatrix} \kappa_{3,3} - \kappa_{4,4} & 0 & 0 & \kappa_{1,4} \\ 0 & 0 & 0 & 0 \\ 0 & 0 & \kappa_{3,3} & 0 \\ \kappa_{4,1} & 0 & 0 & \kappa_{4,4} \end{pmatrix}, \tag{89}$$

and

$$q_L^{(2),L} \propto \begin{pmatrix} \frac{\tilde{\kappa}_{3,3}}{2} - \tilde{\kappa}_{4,4}\csc^2\psi & 0 & 0 & \tilde{\kappa}_{4,1}\csc^2\psi \\ 0 & \zeta & 0 & 0 \\ 0 & 0 & -\tilde{\kappa}_{3,3}\cot^2\psi - \zeta & 0 \\ \tilde{\kappa}_{1,4}\csc^2\psi & 0 & 0 & \tilde{\kappa}_{4,4}\csc^2\psi + \tilde{\kappa}_{3,3}\left(\frac{1}{2} - \csc^2\psi\right) \end{pmatrix}, \tag{90}$$

with

$$\zeta = \tilde{\kappa}_{3,3}\left(\tilde{\kappa}_{4,4}\cot\psi - \frac{1}{2}\right) - \left(\tilde{\kappa}_{4,4}^2 + \tilde{\kappa}_{1,4}\tilde{\kappa}_{4,1} + 1\right)\cot\psi. \tag{91}$$

---

[18]We notice that the condition $\kappa_{3,3} = 0$ is compatible with the fact that the dissipator in (73) is traceless.

[19]We remark that equivalently one can use the jump operators $\ell_{1/L}^{(2)}$ as given in (87) and (88) and $\ell_{1/L}^{(1)}$ to be proportional to $\sigma^-$.

We repeat the same analysis as in the previous section and we obtain that we need to fix the constants to be

$$\kappa_{3,3} = 0\,, \qquad \kappa_{4,4} = -\frac{1}{2}\left(\kappa_{1,4} - \kappa_{4,1}\right)\,, \tag{92}$$

and

$$\tilde{\kappa}_{3,3} = \frac{\csc^2\psi\left(\left(\tilde{\kappa}_{4,4}^2 + \tilde{\kappa}_{1,4}\tilde{\kappa}_{4,1} + 1\right)\sin(2\psi) + \tilde{\kappa}_{1,4} - \tilde{\kappa}_{4,1} - 2\tilde{\kappa}_{4,4}\right)}{2\tilde{\kappa}_{4,4}\cot\psi - 2}\,, \tag{93}$$

$$\tilde{\kappa}_{4,4} = \frac{1}{4}\left(-\csc\psi\sec\psi\sqrt{(\tilde{\kappa}_{1,4}+\tilde{\kappa}_{4,1})^2\sin^2(2\psi)+4} + 2\tilde{\kappa}_{1,4} - 2\tilde{\kappa}_{4,1} - 4\cot(2\psi)\right)\,. \tag{94}$$

We obtain the following expressions for the $\ell$ operators

$$\ell_1^{(1)} = \alpha\,\sigma^+\,, \qquad \ell_1^{(2)} = \beta\,\sigma^-\,, \tag{95}$$

and in the same way

$$\ell_L^{(1)} = \tilde{\alpha}\,\sigma^+\,, \qquad \ell_L^{(2)} = \tilde{\beta}\,\sigma^-\,, \tag{96}$$

where $\alpha$, $\beta$, $\tilde{\alpha}$ and $\tilde{\beta}$ are free constants.

We conclude that, the XX spin chain with magnetic field in the bulk, with boundary terms described by the jump operators (95) and (96) is Yang-Baxter integrable.

**XX spin chain with magnetic field $h = 1$**  We now consider the XX spin chain in magnetic field, when the strength of the magnetic field is $h = 1$. Contrary to the other models, in this case the first derivative of the transfer matrix $t(u)$ does not vanish and generates the dynamics. The analytical expression is

$$\mathbb{Q}_2 = t'(u)|_{u=0} = 2\,q_{Bulk}^{(2)} + q_L^{(2),L} + q_1^{(2),R}\,, \tag{97}$$

where

$$q^{(2),R} = \partial_u K^{(2),R}(u)|_{u=0} = \begin{pmatrix} \kappa_{3,3} - \kappa_{4,4} & 0 & 0 & \kappa_{1,4} \\ 0 & 0 & 0 & 0 \\ 0 & 0 & \kappa_{3,3} & 0 \\ \kappa_{4,1} & 0 & 0 & \kappa_{4,4} \end{pmatrix}\,, \tag{98}$$

and

$$q^{(2),L} = \begin{pmatrix} \tilde{\kappa}_{3,3} - \tilde{\kappa}_{4,4} & 0 & 0 & \tilde{\kappa}_{1,4} \\ 0 & 0 & 0 & 0 \\ 0 & 0 & \tilde{\kappa}_{3,3} & 0 \\ \tilde{\kappa}_{4,1} & 0 & 0 & \tilde{\kappa}_{4,4} \end{pmatrix}\,. \tag{99}$$

The expression for the boundaries coincide with the ones in (89), so in order to bring the solution to a Lindbladian form, we have to require the condition (92) for the coefficients (both $\kappa$s and $\tilde{\kappa}$s) and the expressions for the $\ell$ operators are (95) and (96).

It is worth to mention that for this model, an interesting phenomenon happens. The first non-trivial conserved charge is generated by taking the first derivative of the transfer matrix, however, the analytical expression does not coincide with (26) given by Sklyanin in [49]. The reason is that, in Sklyanin's derivation, he assumes that taking the limit of the product of matrices as the spectral parameter approaches zero is equivalent to taking the limit of each term in the product individually. However, there can be cases, such as the one we are considering, where this assumption does not hold. In our case, one factor diverges in the limit. Taking the product first regularizes the result. We comment further on this statement in Appendix D.

The terms contributing to the left boundary are the ones corresponding to taking the derivative of the terms $R_{0L}(u)$ (both in $T(u)$ and $\hat{T}(u)$) and $K^L(u)$.

# 6 Conclusion

In this work, we constructed quantum circuits on a qubit ladder where the elementary quantum gate in the bulk is a tensor product of a pair of two qubit gates. We investigate whether the quantum gates at the left and right boundaries may exhibit a non-factorized expression while preserving integrability. We obtain that non-factorized solutions exist only if the bulk model is the free fermion XX spin chain. In contrast, for the other models analyzed, namely the XXX and XXZ spin chains, all solutions are factorized. For the free fermion model, we examine the continuous time dynamics and find that the boundary terms can be expressed as a Lindbladian evolution for a specific choice of free parameters. This means that the operators at the left and right boundaries of the spin chain act as sources for particle injection or removal. This result implies that the Yang-Baxter integrability property of a model, both in the bulk and at the boundaries, imposes stricter conditions than those required to find the analytical expression of the NESS of the model.

The free fermionic property of the model in the bulk seems to be also a necessary condition to guarantee the Yang-Baxter integrability of the Lindbladian superoperator with dissipator terms acting on the bulk of the spin chain, [19, 50, 51].

This result is somehow reminiscent to a similar phenomenon in integrable quantum field theory (IQFT) with extended defect lines or with an impurity,[20] [52, 53]. In IQFT, there exist two types of integrable defects, [52]: topological (purely trasmissive) and non-topological (transmissive and reflective) defects. By using the so-called "folding trick", the defects can be related to the boundary. In [53], it was shown that the only theories that allows non-topological integrable defects are the free theories (free fermion or free boson). Theories characterized by non-topological defects are related to the factorizability of the boundary S-matrices.

Many interesting open questions need to be addressed. One may introduce the coupling between the two free fermionic (XX) chains either in the form of a Hubbard interaction in the unitary context or dephasing in the open systems framework, and again ask the question about factorizability or Lindbladian form of the boundary terms. Furthermore, we observed that the circuit of Fig. 10 where the gate is the XX model, corresponds to a finite free fermionic model with open boundaries and an interacting impurity in the middle. For a particular choice of the parameters of the impurity, one has $U(1)$ symmetry which allows to study impurity related transport properties of the model. Furthermore, since all the models constructed are Yang-Baxter integrable, it is also possible to obtain their spectrum via one of the Bethe Ansatz techniques. Another interesting direction is to study factorizability of $K$-matrices for higher local Hilbert space dimension.

## Acknowledgments

We would like to thank Z. Bajnok, M. de Leeuw, E. Ilievski, Y. Kasim, A. Klümper, R. Nepomechie, V. Popkov, A. L. Retore, A. Torrielli, L. Zadnik, M. Žnidarič for useful discussions. We also thank A. L. Retore for useful comments on the manuscript. C.P. would like to thank P. Vieira for the kind hospitality at ICTP SAIFR in São Paulo during the last stage of this work.

**Funding information** We acknowledge funding from the European Union HORIZON-CL4-2022-QUANTUM-02-SGA through PASQuanS2.1 (Grant Agreement No. 101113690),

---

[20]We thank Z. Bajnok for pointing out the similarity between the two results.

European Research Council (ERC) through Advanced grant QUEST (Grant Agreement No. 101096208), as well as the Slovenian Research and Innovation agency (ARIS) through the Program P1-0402.

## A  Factorized solutions: Motivation

Here we motivate why if we start from two solutions $k^R$ and $\tilde{k}^R$ of the boundary Yang-Baxter equation (10) for an R-matrix $r(u)$ of a spin 1/2 chain, then their tensor product is a solution in the enlarged space.

The $R$ and $K$ matrices we consider are

$$R_{12,34}(u) = r_{13}(u)r_{24}(-u)\,, \qquad K^R_{12}(u) = k^R_1(u)\tilde{k}^R_2(-u)\,. \tag{A.1}$$

We write the boundary YB equation[21] (10) in the enlarged space, by making the identification $1 = 12$ and $2 = 34$. We obtain

$$r_{13}(u-v)r_{24}(v-u)k_1(u)\tilde{k}_2(-u)r_{31}(u+v)r_{42}(-u-v)k_3(v)\tilde{k}_4(-v)$$
$$= k_3(v)\tilde{k}_4(-v)r_{13}(u+v)r_{24}(-u-v)k_1(u)\tilde{k}_2(-u)r_{31}(u-v)r_{42}(v-u)\,. \tag{A.2}$$

We multiply by $P_{23}$ both members from left and right, and we consider that matrices acting on different spaces commute, we get

$$\left(r_{12}(u-v)k_1(u)r_{21}(u+v)k_2(v)\right)\left(r_{34}(v-u)\tilde{k}_3(-u)r_{43}(-u-v)\tilde{k}_4(-v)\right)$$
$$= \left(k_2(v)r_{12}(u+v)k_1(u)r_{21}(u-v)\right)\left(\tilde{k}_4(-v)r_{34}(-u-v)\tilde{k}_3(-u)r_{43}(v-u)\right)\,. \tag{A.3}$$

We can easily recognize the two boundary YBE for each of the two spin 1/2 chains. This also explains why the free constants in $k^R(u)$ (we labelled them $\kappa$) and $\tilde{k}^R$ (labelled as $\tilde{\kappa}$) are independent. In other words, we proved that one trivial solution of $K(u)$ in the enlarged space can be obtained by taking the tensor product of two solutions of the boundary Yang-Baxter for the spin 1/2 chain.

## B  Twisted boundary conditions

We briefly consider here an intermediate case between open and periodic boundary condition, the twisted or quasi-periodic boundary condition. The transfer matrix takes the expression, [20], [38]

$$t(u, \mathbf{w}) = \text{tr}_0 G_0 R_{0L}(u-w_L)\cdots R_{02}(u-w_2)R_{01}(u-w_1)\,, \tag{B.1}$$

where the twist matrix $G$ satisfies

$$[R_{i,j}(u), G\otimes G] = 0\,. \tag{B.2}$$

In the circuit language, one can fix the inhomogeneities as

$$w_1 = w_3 = w_5 = \cdots = w_{L-1} = -\kappa\,, \qquad w_2 = w_4 = \cdots = w_L = \kappa\,, \tag{B.3}$$

and hence

$$t(u) = \text{tr}_0 G_0 R_{0L}(u-\kappa)\cdots R_{02}(u-\kappa)R_{01}(u+\kappa)\,. \tag{B.4}$$

---

[21]For simplicity we omit the superscript $R$ for the right $K$-matrix.

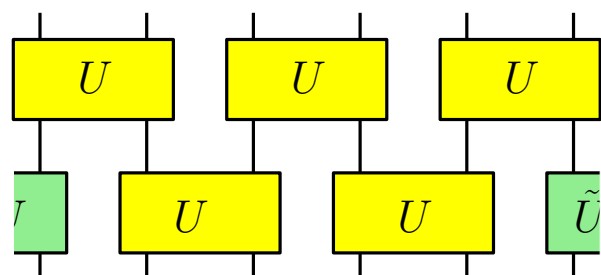

Figure 11: Quantum circuit from twisted boundary condition. The gate $\tilde{U}$ is related to $U$ as $\tilde{U}_{i,j} = G_i^{-1} U_{i,j} G_i$.

Evaluating the transfer matrix at $u = \pm\kappa$,

$$t(\kappa) = P_{1,L} P_{1,L-1} \cdots P_{1,3} P_{1,2} U_{2,3} \cdots U_{L-2,L-1} U_{L,1} G_L \,, \tag{B.5}$$

$$t(-\kappa) = P_{1,L} P_{1,L-1} \cdots P_{1,3} P_{1,2} G_L U_{L-1,L}^{-1} \cdots U_{3,4}^{-1} U_{1,2}^{-1} \,. \tag{B.6}$$

We can define the dynamic as

$$M = t(-\kappa)^{-1} t(\kappa) = U_{1,2} U_{3,4} \cdots U_{L-1,L} U_{2,3} U_{4,5} \cdots U_{L-2,L-1} G_L U_{L,1} G_L^{-1} \,, \tag{B.7}$$

and recognize the two steps in the evolution

$$\mathbb{U}_1 = \left( \prod_{k=1}^{\frac{L}{2}} U_{2k-1,2k} \right) \,, \qquad \mathbb{U}_2 = \left( \prod_{k=1}^{\frac{L}{2}-1} U_{2k,2k+1} \right) G_L^{-1} U_{L,1} G_L \,, \tag{B.8}$$

which corresponds to the circuit of Figure 11.

This type of boundary conditions are not independent from the open boundary conditions, but they can be related to it, as explained in [3].

Interestingly, from the point of view of quantum circuits, a dynamic similar to Figure 11 can be obtained from the case of open boundary condition where the length of the spin chain is even, [14]. It is easy to see that for that case

$$t(u) = \mathrm{tr}_0\Big( K_0^L(u) R_{0L}(u-\kappa) \cdots R_{02}(u-\kappa) R_{01}(u+\kappa) K_0^R(u) R_{10}(u-\kappa) R_{20}(u+\kappa) \cdots R_{L0}(u+\kappa) \Big), \tag{B.9}$$

$$t(\kappa) = U_{12} U_{34} \cdots U_{L-1,L} K_1^L(\kappa) U_{23} U_{45} \cdots U_{L-1,L-2} \mathrm{tr}_0\Big( K_0^R(\kappa) U_{0L} \Big), \tag{B.10}$$

which is equivalent to the dynamics of Figure 11 with the gate $\tilde{U}_{i,j} = K_i^L(\kappa) \mathrm{tr}_0\Big( K_0^R(\kappa) U_{0j} \Big)$.

# C  Allowed transformations

It is known that given an $R$-matrix solution of the Yang-Baxter equation (2), certain transformations can be applied to it, ensuring it remains a solution. These transformations are described in details in [54–56]. Here, we analyze what happens to the $K$-matrices if we apply these transformations. In what follow, we first write how each transformation acts on the $R$-matrix and then on the $K^{L/R}$. We omit the superscript $L/R$ since the transformations act on the same way on both the $K$-matrices.

**Local basis transformation**

$$R_{12} \rightarrow A_1 A_2 R_{12} A_1^{-1} A_2^{-1}, \tag{C.1}$$

$$K_1 \rightarrow A_1^{-1} K_1 A_1. \tag{C.2}$$

**Twist**

$$R_{12} \rightarrow A_2 R_{12} A_1^{-1}, \qquad [R_{12}, A_1 A_2] = 0, \tag{C.3}$$

$$K_1 \rightarrow A_1^{-1} K_1 A_1. \tag{C.4}$$

**Normalization**

$$R_{12}(u) \rightarrow f(u) R_{12}(u), \qquad K_1(u) \rightarrow g(u) K_1(u). \tag{C.5}$$

**Reparametrization**

$$u \rightarrow h(u), \quad \text{such that} \quad h(u-v) = h(u) - h(v), \tag{C.6}$$

where $A$ is an invertible matrix, $f(u), g(u)$ are arbitrary functions and $h(u)$ is an arbitrary functions such that $h(u-v) = h(u) - h(v)$.

# D  Explicit calculation of the first two derivatives of $t(u)$

We start from the expression of the transfer matrix with all the inhomogeneities set to 0 and we give the analytical expressions of the higher order derivatives. We use the expression (12) for the double raw transfer matrix. By omitting the dependence on the spectral parameter, this is

$$t(u) = \text{tr}_0 K_0^L R_{0L} \cdots R_{02} R_{01} K_0^R R_{01} R_{02} \cdots R_{0L}, \tag{D.1}$$

where we restrict to $R$-matrix that satisfies the symmetricity and unitarity properties.

We consider the assumption that

$$\lim_{u \to 0} A_1(u) A_2(u) \cdots A_{2L+2}(u) = \lim_{u \to 0} A_1(u) \lim_{u \to 0} A_2(u) \cdots \lim_{u \to 0} A_{2L+2}(u), \tag{D.2}$$

where $A_i(u)$ are the matrices entering in the transfer matrix definition. We remark that this is also the assumption used by Sklyanin in [49] and that, for the case of the XX spin chain with magnetic field $h = 1$ that we treat in section 5.2.3, the assumption does not hold. We motivate it in section D.1.

We consider the following boundary conditions

$$R_{ij}(0) = P_{ij}, \qquad K_i^R(0) = \mathbb{I}, \qquad \partial_u R_{ij}(u)|_{u \to 0} = P_{ij} h_{ij}. \tag{D.3}$$

For simplicity, in this appendix we refer to $q_{i,j}^{(2)} = h_{i,j}$ as the range 2 conserved density. For the models treated in the main text, this is the superoperator $\mathcal{L}$ given in (71).

## D.1  First derivative $t'(0)$

By direct calculation, we can derive

$$t'(0) = 2\text{tr}_0 K_0^L(0) H_{Bulk} + \text{tr}_0 K_0^L(0) K_1^{R'}(0) + 2\text{tr}_0 K_0^L(0) h_{L0} + \left(\text{tr}_0 K_0^{L'}(0)\right) \mathbb{I}, \tag{D.4}$$

with

$$H_{Bulk} = \sum_{i=1}^{L-1} h_{i,i+1}\,. \tag{D.5}$$

The last term is proportional to the identity matrix, so we can write the charge generated by

$$H_{tot} = H_{Bulk} + H_1^R + H_L^L\,, \tag{D.6}$$

with

$$H_{Bulk} = \sum_{i=1}^{L-1} h_{i,i+1}\,, \qquad H_1^R = \frac{1}{2}K_1^{R'}(0)\,, \qquad H_L^L = \frac{\mathrm{tr}_0 K_0^L(0)h_{L0}}{\mathrm{tr}_0 K_0^L(0)}\,. \tag{D.7}$$

This prescription is very general, however for the models under consideration $\mathrm{tr}_0 K_0^L(0) = 0$, so to construct the correct range 2 conserved charge, one have to take higher order derivatives.

For the XX spin chain with magnetic field $h = 1$ that we treat in section 5.2.3, the first derivative of the transfer matrix generates the dynamics. For that model, an interesting phenomenon happens and the expression (D.4) does not hold. In fact, the term $2\mathrm{tr}_0 K_0^L(0)h_{L0}$ that comes from taking the derivative of $R_{0L}(u)$ ($\lim_{u\to 0}\mathrm{tr}_0 K_0^L(u)R_{0L}'(u)R_{0,L-1}(u)\dots$) is divergent. However, adding it to the terms coming from $\lim_{u\to 0}\mathrm{tr}_0 K_0^{L'}(u)R_{0L}(u)R_{0,L-1}(u)\dots$ cancels the divergent terms. Since (D.3) holds, we may be tempted to conclude that for the term added, the following identity holds

$$\lim_{u\to 0}\mathrm{tr}_0 K_0^{L'}(u)R_{0L}(u)R_{0,L-1}(u)\cdots = \lim_{u\to 0}\mathrm{tr}_0 K_0^{L'}(u)\,. \tag{D.8}$$

This is not correct for this model, in fact the r.h.s. is divergent, while the l.h.s. is finite. To obtain the correct result, one should first add the terms coming from the derivatives of $R_{0L}$ and $K_0^L$ and only afterward take the limit.

## D.2 Second derivative $t''(0)$

We consider the second derivative of the transfer matrix. By a lengthy but straightforward calculation,[22] one can obtain the following expression

$$\begin{aligned}
t''(0) = {}& 2\Big(\mathrm{tr}_0 K_0^{L'}(0) + 2\mathrm{tr}_0 K_0^L(0)h_{0,L}\Big)K_1^{R'}(0) + 4\mathrm{tr}_0 K_0^{L'}(0)H_{Bulk} + 4\{\mathrm{tr}_0 K_0^L(0)h_{0,L}, H_{Bulk}\} \\
& + 2\Big(\mathrm{tr}_0 K_0^L(0)R_{0,L}{}''(0) + 2\mathrm{tr}_0 K_0^{L'}(0)h_{0,L} + \mathrm{tr}_0 K_0^L(0)h_{0,L}^2\Big) + \mathrm{tr}(K_L{}''(0))\,\mathbb{I} \\
& + 2(\mathrm{tr}_0 K_0^L(0))\Big(\sum_i (h_{i,i+1})^2 + \{H_{Bulk}, K_1^{R'}(0)\} + \frac{1}{2}K_{R,1}{}''(0) + \sum_i \check{R}_{i,i+1}''(0)\Big) \\
& + 2\,\mathrm{tr}_0 K_0^L(0)\sum_{\substack{i,j=1 \\ i\neq j}}^{L-2} \{h_{i,i+1}, h_{j,j+1}\}\,. \tag{D.9}
\end{aligned}$$

We did not find this explicit computation in the literature. In [57], the authors gave the expression of $t''(u)$ for the case with $\mathrm{tr}_0 K_0^L(0)h_{L,0} \propto \mathbb{I}$ and $\mathrm{tr}_0 K_0^L(0) = 0$. We checked that, under these assumptions, our expressions reduce to theirs. In that case, the expression of the second derivative, takes the expression (D.6). However, in the absence of these assumptions, the charge cannot be written in a form similar to (D.6). The terms acting in the boundary will have interaction range 2 and the ones in the bulk range 3.

---

[22]We performed this calculation with the software Mathematica, by implementing the non-commutative product and calculating the derivative for a chain of finite length and we extract the general expression.

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
