# Peer review of "Integrability of open boundary driven quantum circuits"

_SciPost Physics, doi:SciPost Phys. 18, 027 (2025)_

## Round 2 · Referee Report · Anonymous (Referee 1) · 2024-9-4

Strengths
1-Interesting new results by classifying quantum circuits with open boundaries
2-Clearly written
Weaknesses
1-Not too much of the physical properties of the new models are investigated and it's not clear if they give some interesting physics.
Report
This paper looks for integrable quantum circuits with open boundary conditions. The authors consider a set-up where they consider the evolution of a chain of spin 1/2 particles. They consider different types of bulk unitary gates, in particular cases with and without interactions. They then proceed to solve the reflection equation in these different cases and see how they fit into a 2 layer brickwork pattern and see if they factorise.
The paper is very clearly written and gives a nice introduction to all the concepts that are needed to understand the results. I think the results are relevant and could potentially describe some interesting physical models. I recommend this paper for publication.
Requested changes
- Can the authors comment on they expect the models that they find to be exactly solvable by some sort of Bethe ansatz?
- There is a case between periodic boundary conditions and open boundary conditions. These are quasi-periodic or twisted boundary conditions. In this case you need a matrix G such that [R,GG]=0. Then you can insert G into the transfer matrix and you find that G describes the relation between the last and first site. Can this case be obtained from the results in the paper or would it require a separate analysis. Would this case be interesting?
Recommendation
Publish (easily meets expectations and criteria for this Journal; among top 50%)

---

## Round 2 · Referee Report · Anonymous (Referee 3) · 2024-9-26

(Invited Report)- Cite as: Anonymous, Report on arXiv:2406.12695v2, delivered 2024-09-26, doi: 10.21468/SciPost.Report.9810
Report
In this paper, the authors investigated a special type of integrable quantum circuits with open boundaries. The bulk fundamental gate is the tensor product of a pair of two qubit gates which satisfy the Yang-Baxter equation. The gates at the boundary is required to satisfy Sklyanin's reflection algebra, or boundary Yang-Baxter equation so that the whole system is integrable. They ask the question whether the boundary K-matrix can have a non-factorized solution. By solving the reflection algebra explicitly, they show that non-factorized solutions only exist when the bulk gate corresponds to the $R$-matrix of the free XX spin chain. In this case, and for specific choices of boundary parameters, the circuit system can be viewed as a discretized Lindbladian evolution system.
This work is interesting enough to be published on SciPost. The paper is clearly written and reviewed proper amount of contents. I only have some minor comments.
1. I think some brief review of integrable Lindblad evolution system could be helpful for the reader. The reason is that without this context or background, it is slightly difficult to appreciate why the authors consider the double copy system and investigate the specific question of factorizability of the boundary K-matrix.
2. The conclusion of this paper, namely only the free bulk theory can give rise to the non-factorized boundary K-matrix reminds me some very similar results in the context of integrable quantum field theory (IQFT). For an IQFT, one can consider defects and there is a notion of an integrable defect, defined Delfino, Mussardo and Semonetti (see hep-th/9409076). There are two types of integrable defects, which are the topological and non-topological defects. The topological defects are purely transmissive while the non-topological ones are simultaneously transmissive and reflective. In the work of Catro-Alvaredo, Fring and Gohmann (see hep-th/0201142), they showed that the only theories that allows non-topological integrable defects are the free theories (free fermion or free boson). It is known that the defects can be related to the boundary by the so-called “folding trick”. Roughly speaking, “folding” the theory along the defect gives a boundary. It seems to me that the folding trick gives two copies of the bulk theory naturally and the distinction between topological and non-topological defects are related to the factorizability of the boundary S-matrices. The two conclusions are very similar and might actually be related in some cases. It would be interesting if the authors could make some comments on this.
Requested changes
1. Give a brief review on integrable Lindblad evolution system;
2. Comment on the point which I raised in the report (this is not compulsory).
Recommendation
Ask for minor revision
Strengths
1. Timely and interesting topic
2. Developing a general framework
3. Clear and well-written exposition
Weaknesses
None
Report
The paper develops a framework for constructing Yang-Baxter integrable open systems with Lindblad jump operators located at the boundary. This framework is especially powerful given its generality, and in my view, the most exciting conclusion is that all such systems correspond to free fermionic dynamics in the bulk. I have no hesitation in recommending this work for publication in Scipost Physics.
Requested changes
None
Recommendation
Publish (easily meets expectations and criteria for this Journal; among top 50%)

---

## Round 3 · Referee Report · Anonymous (Referee 1) · 2024-11-13

Report

I am happy with the current version and recommend it for publication now.

Recommendation

Publish (easily meets expectations and criteria for this Journal; among top 50%)

---

## Round 3 · Referee Report · Anonymous (Referee 3) · 2024-11-13

Report

In the revised version, the authors have addressed the comments in my previous report. I gladly recommend it to be published now.

Recommendation

Publish (meets expectations and criteria for this Journal)

---

## Round 3 · Author Response

We thank the anonymous referee for the useful comments. We provide an answer to the questions posed in the report.
Report 1
1. All the models treated in this paper are constructed to be Yang-Baxter integrable, and hence, various integrability techniques (such as the Bethe ansatz or Baxter Q-operators) can be used to solve them. For instance, in certain cases, the algebraic Bethe ansatz can be applied. We refer to Sklyanin's original paper for the case of diagonal K-matrices. Following this comment, we have also added additional references in the manuscript for the cases with non-diagonal boundaries and some examples where Bethe ansatz techniques were applied in the context of quantum circuits.
2. In the context of quantum circuits, the dynamics generated by this type of transfer matrix is reminiscent of the case with open boundary conditions, where the length of the spin chain is even. As mentioned by Sklyanin in the original paper, if a specific representation of the K-matrix is chosen, the double-row transfer matrix reduces to a transfer matrix with quasi-periodic boundary conditions. In our paper, we use a different representation, so this solution is not included in our analysis. However, we found this question interesting and have added Appendix B to clarify the dynamics arising from this case. We have checked that for our models, the twist G also factorizes.
Report 2
No changes or questions asked
Report 3
1. This was indeed an important point. We have expanded the paragraph “Motivation of the paper” at the end of page 4. Additional and more precise motivations are given in section 5.
2. We thank the referee for the comment. We were already aware of this analogy and the relevant references, as it was previously pointed out to us by Z. Bajnok. Based on the referee's suggestion, we have expanded the sentence in the conclusion. However, we did not explore this topic further, though we are open to further discussion if the referee would like to suggest additional cases to investigate.

---

## Round 3 · List of Changes

For report 1:
1. We have answered this question at the end of section 2.2.1. To clarify, we also cited cited:
- J. Phys. A: Math. Gen. 21 2375 for Bethe ansatz with diagonal boundary
- 2012.08367 for Bethe ansatz with non-diagonal boundary
- 2408.474 Bethe ansatz for periodic quantum circuits
2. In 2.2 we added the sentence: “For completeness, in Appendix B, we also consider twisted boundary conditions.”
We have added appendix B. To clarify, we added the references: 2207.14193 and 1009.4118
For Report 3:
1. We changed the title of a paragraph in the Introduction: “Integrability in boundary driven diffusive Lindbladian systems”.
We have expanded the paragraph “Motivation of the paper” at the end of page 4.
We added in 3.1 “For the reason that we briefly mentioned in the introduction and that will be clarified in the following,”
2. We have expanded the sentence in the conclusion about IQFT.

---

## Editorial Decision

published